

**Rapid Landslide Mapping During the 2023 Emilia-Romagna Disaster: Assessing Automated**
**Approaches with Limited Training Data**
*Nicola Dal Seno[1]\*, Giuseppe Ciccarese [1], Davide Evangelista[2], Elena Loli Piccolomini[2], Alessandro*
*Corsini[3], Matteo Berti[1]*
[1] Department of Biological, Geological, and Environmental Sciences (BiGeA), University of
Bologna, via Zamboni 67, Bologna, Italy
[2] Department of Informatics: Science and Engineering (DISI), University of Bologna, Bologna, Italy
[3] Department of Chemical and Geological Sciences, University of Modena, Modena, Italy
\*Corresponding author
email: nicola.dalseno@unibo.it
full postal address: Via Zamboni, 67 - 40126 Bologna
**Abstract**
The catastrophic rainfall events of May 2023 in the Emilia-Romagna region, Italy triggered more than
80.000 landslides (https://doi.org/10.5281/zenodo.13742643, Pizziolo et al., 2024) and placed
extraordinary demands on emergency response systems. One of the most critical tasks during the
emergency was mapping the landslides caused by the event, a process carried out manually that
required substantial time and effort (Berti et al., 2024).This study explores the potential of automated
landslide mapping to support rapid disaster response, evaluating two deep learning models, U-Net
and SegFormer, under realistic emergency constraints, including minimal training data. The models
were trained exclusively in one affected municipality area (Casola Valsenio, 84 km$^2$) and tested on
three additional municipalities (Predappio, 91 km$^2$; Modigliana, 101 km$^2$; Brisighella, 194 km$^2$) with
varying geological settings. To reflect a range of operational scenarios, we tested seven combinations
of input layers, progressively increasing in complexity from post-event Sentinel-2 imagery alone to
full integration of high-resolution aerial imagery, NDVI change maps, and slope data.
Results show that both models achieved comparable segmentation performance, with SegFormer
displaying greater robustness to variations in input layers and geological conditions, while U-Net was
more sensitive but occasionally more accurate with rich inputs. Both models successfully identified
landslides, but encountered difficulties in shadowed zones, cultivated fields, and geologically distinct
terrains. A key limitation emerged in Brisighella FAA, a sector dominated by Blue Clay formations,
where poor generalization was traced to the lack of lithological diversity in the training data,
highlighting the need for geologically balanced training datasets.
Despite these challenges, the study confirms the operational value of automated mapping as a first-
pass tool. Both models delivered accurate baseline maps that could support manual validation and
prioritization in time-critical scenarios. The findings suggest that AI-based mapping could become a
key component of emergency management protocols. While manual revision remains essential, these
tools offer a scalable, time-efficient solution to the increasing demand for rapid and spatially detailed
hazard information.




## 1. Introduction

Rapid and accurate landslide mapping over large areas is a challenging task, particularly in the context
of regional-scale rainfall or earthquake events (Iverson et al. 2015; Casagli et al. 2016; Robinson et
al., 2017; Holbling et al. 2017; Mondini et al., 2021). This task is crucial for implementing timely and
effective response measures, assessing the extent of impact, and planning for recovery and mitigation
efforts. The complexities of rapid landslide mapping vary depending on several factors including the
extent of the affected area, the availability of cloud-free imagery, the precision needed for the
mappings, and the difficulties of detecting landslides due to shadows, vegetation cover, or weak
geomorphic evidences (Mondini et al. 2019; Amatya et al., 2023). Typically, these complexities are
addressed by expert geologists through the visual interpretation of satellite or aerial imagery (Guzzetti
et al., 2012; Scaioni et al., 2014; Ferrario and Livio 2023). These experts are skilled at identifying
subtle variations in the terrain and can integrate complex contextual knowledge, including field
reports, personal accounts, and specific geological information, to create highly accurate landslide
maps. However, manual mapping is both time-consuming and inherently subjective, which are
significant limitations during a crisis (Novellino et al. 2024).
Automated landslide recognition techniques using convolutional neural networks (CNN) algorithms
are emerging as promising alternatives to manual methods (Sameen and Pradhan 2019, Chen et al.
2018, Ye et al. 2019, Tang et al. 2022, Ji 2020). Recent studies have demonstrated their effectiveness
in real-world cases. For instance, Meena et al. (2021) applied a deep learning approach for rapid
landslide mapping in India following extreme monsoon rainfall, showcasing the potential of these
techniques in post-disaster conditions. Prakash et al. (2021) introduced a new strategy using a
generalized convolutional neural network, aiming to develop a more versatile method applicable
across various geographical contexts. Building on this, Prakash and Manconi (2021) demonstrated
the practical utility of these methods by rapidly mapping landslides triggered by a severe storm event.
These studies emphasize the growing importance of CNNs in automated landslide mapping and their
potential for practical implementation in disaster management. However, despite their proven
effectiveness, a notable gap remains between theoretical applications and their real-world deployment
during emergencies. Typically, these studies employ a large train-to-validation ratio (usually 80:20
or 70:30; Meena et al 2023), which does not mirror the limited data availability for training models
in real-time crisis situations. Furthermore, the training process frequently does not account for the
variety of landslide types and the spectrum of geological conditions present in actual situations. This
oversight is critical, as the complexity of natural environments play a crucial role in the effective
application of these technologies in emergency contexts.
In this study, we utilize the landslide inventory from the May 2023 crisis in the Emilia-Romagna
region of Italy (Berti et al. 2025) to assess the effectiveness and limitations of automated mapping
algorithms in a practical setting. We specifically focus on the performance of the U-Net and
SegFormer algorithms in identifying landslides triggered by the event, despite the challenges of very
limited training data. The study also considers the variety of landslide types and geological conditions
prevalent in the area, which impacted the disaster response. The main goal of this research is to
determine whether advanced deep-learning methods can effectively replace manual mapping in



managing large-scale landslide disasters, thus bridging the gap between academic research and real-
world application in emergency management.

## 2. The Romagna May 2023 disaster

### 2.1 Overview of the disaster

In May 2023, the Emilia-Romagna region of Italy was hit by unprecedented meteorological events,
as detailed by Berti et al. (2025) and Foraci et al. (2023). Two major rainfall episodes, from May 1-3
and May 16-17, led to severe floods and landslides primarily affecting the eastern part of the region
(Fig. 1a). The first event (May 1-3) was driven by a stationary low-pressure system which delivered
over 200 mm in 48 hours. The second event (May 16-17), featured a similar low-pressure system
enhanced by bora winds and moist air from the Mediterranean, resulting in rainfall exceeding 250
mm in the same duration. The cumulative impact of these episodes resulted in more than 500 mm of
rainfall over two weeks, nearly half the region's annual climatic precipitation (Fig. 1a). The return
period for these combined rainfall events was estimated to exceed 1000 years (Brath et al. 2023),
highlighting the unprecedented nature of this occurrence. The two cyclonic events led to extensive
flooding, submerging 540 km² of land, and triggered more than 80,000 landslides across
approximately 1000 km² (Fig. 1b). This widespread devastation impacted thousands of buildings,
roads, and other infrastructure, with the estimated damages amounting to approximately 9 billion
euros.
Immediately after the disaster, we initiated the development of a detailed inventory map of the
landslides caused by the event. This map was manually created using high-resolution aerial imagery
taken only one week following the second rainfall event. The images, which include four bands
(RGB+NIR) and have a resolution of 0.2 m, allowed for the accurate identification and classification
of the landslides. The data derived from this mapping effort are described in Berti et al. (2025) and
are available for public download in shapefile format (DOI: 10.5281/zenodo.13742643, Pizziolo et
al., 2024).

### 2.2 Motivation of the work

The visual identification of the landslides from May 2023 proved to be a challenging and time-
intensive process. It took six months for twelve expert geologists to identify and delineate all the
landslides within the 1000 km² area affected by the disaster. Despite these challenges, this approach
was deemed the best available method at the time to meet the Civil Protection agency's specific
requirements for accuracy, precision, and oversight during the emergency response coordination.
As this manual mapping progressed, we explored various automated mapping techniques to speed up
the process and reduce the workload. Initial trials using NDVI methods and U-Net in a specific sample
area produced promising results, demonstrating that deep-learning models can effectively replicate
manual mapping efforts (Berti et al., 2024). However, these preliminary analyses were conducted in
a test area (the Casola Valsenio municipality, Fig. 1), which covers only about 10% of the total area
impacted by the event. Consequently, several critical questions remain unanswered: i) Can automated
models trained on a small area be effectively used for rapid mapping of larger regions? ii) How do





models trained in specific geological conditions perform across different geological settings? iii)
Could more sophisticated deep-learning methods surpass U-Net in improving outcomes?
Addressing these questions is crucial to determine whether emerging automated methods can enhance
the efficiency of landslide mapping in future crisis scenarios, thereby potentially reducing critical
response times. In this study, we simulate landslide mapping during a crisis, taking into account the
limited data typically available and the accessible information under such conditions. This simulation
is grounded in our firsthand experience of the real emergency during the May 2023 event, which
provided us with valuable insights into the practical challenges and data limitations encountered
during such crises.

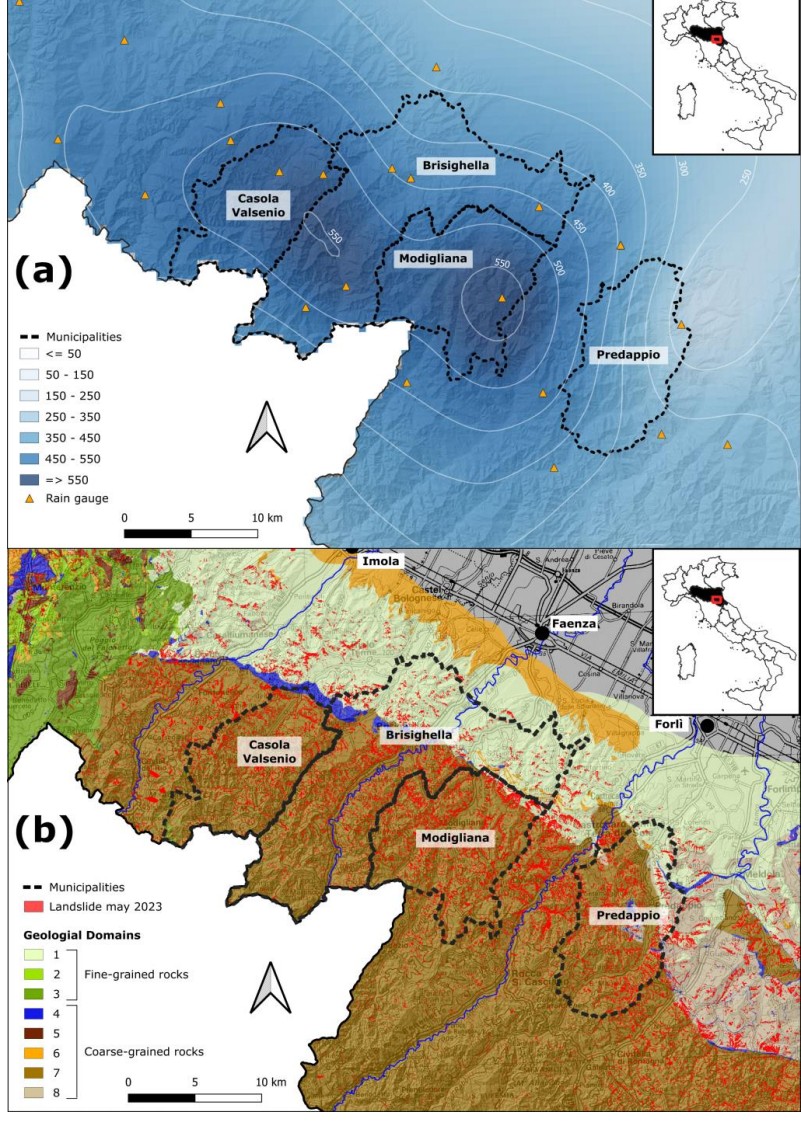


**Figure 1.** Overview of the Effects of the Extreme Rainfall Event in Romagna, May 2023.



**a**) Cumulative precipitation map from May 1 to May 17, 2023. **b**) Manually mapped landslide distribution across the study areas, categorized by geological formations as per the classification by Berti et al. (2025).

## 3. Study area

Automated methods for landslide mapping were applied in the four municipalities shown in Figure 2. These areas, situated in the eastern part of the Emilia-Romagna region (Fig. 1), were significantly affected by the meteorological event of May 2023. The Casola Valsenio area was chosen for training models, whereas Brisighella, Modigliana, and Predappio served as validation sites. In their study, Berti et al. (2024) used Casola Valsenio to train a conventional U-Net model, and compared the results with those obtained using the traditional NDVI change detection method. We opted to continue using this area for training to ensure consistency with previous research. The other three areas were selected because together they encompass approximately 35% of the regions most impacted by landslides and represent about 25% of the recorded 80,997 landslide events. Moreover, these municipalities exhibit varied geological conditions, with both similarities and differences to Casola Valsenio, thereby offering a comprehensive test of the models' adaptability to diverse terrains and landforms.

To represent geological variability in a way that is both meaningful and operational for the analysis, we adopted the lithological classification introduced by Berti et al. (2025), which groups the over 600 geological formations of the Emilia-Romagna region into eight lithological units (Fig. 1b). This classification combines lithological characteristics with structural domains, acknowledging that the same rock type may exhibit different mechanical properties depending on its tectonic setting. Units 1 to 3 consist of fine-grained materials (e.g., clays, marls, tectonized shales), while units 4 to 8 mainly comprise coarse-grained rocks (e.g., sandstones, flysch, conglomerates). This simplified scheme captures key differences in the weathering behavior and mechanical response of the rock masses, which are known to influence the type and frequency of landslides. Fine-grained units generally produce cohesive soils prone to earth slides and flows, whereas coarse-grained units generate granular soils more susceptible to debris slides and debris flows. This categorization aligns with the "earth" and "debris" material types defined in the Cruden and Varnes (1996) landslide classification and is used throughout this study to interpret landslide patterns in relation to geological setting.



171

172

**Table 1.** Simplified classification of the geological formations in the Emilia-Romagna region into eight litho-structural units, adapted from Berti et al. (2025). The classification is based on lithological composition and structural domain. Units 1 to 3 predominantly include fine-grained rocks, while units 4 to 8 are composed mainly of coarse-grained lithologies.

| Unit ID | Lithology | Domain | Structural position | Geological Age |
|---|---|---|---|---|
| 1 | Clays, silty clays, and marly clays | Padano-Adriatic | Outer Foredeep | Pliocene to Pleistocene |
| 2 | Marls and marly clays | Epiligurian | Wedge-top basins | Oligocene to Miocene |
| 3 | Clay shales, clay breccias, tectonized clays, olistostromes | Ligurian | Accretionary wedge | Cretaceous to Eocene |
| 4 | Massive rocks: basalts, serpentines, limestones, arenites | Ligurian, Epiligurian | Accretionary wedge Wedge-top basins | Cretaceous to Miocene |
| 5 | Flysch rocks made of rhythmic alternations of sandstones, limestones, pelites, and shales | Ligurian, Epiligurian | Accretionary wedge Wedge-top basins | Cretaceous to Eocene |
| 6 | Weakly cemented sandstones and conglomerates | Padano-Adriatic | Outer Foredeep | Pliocene to Pleistocene |
| 7 | Flysch rocks made of rhythmic alternations of sandstones and pelites | Tuscan-Umbrian | Inner Foredeep | Miocene |
| 8 | Weakly cemented sandstones with interbedded pelitic layers | Padano-Adriatic | Outer Foredeep | Pliocene to Pleistocene |

177





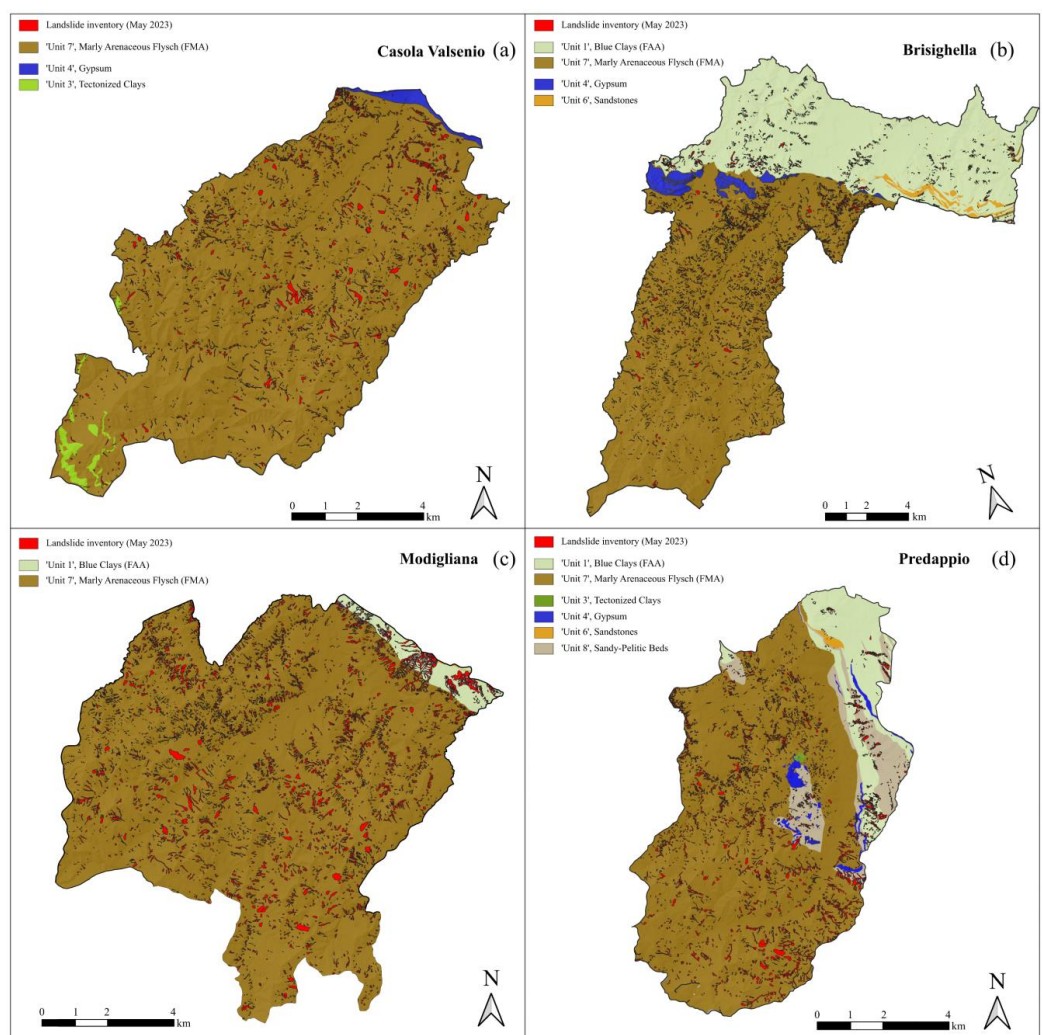

**Figure 2.** Schematic geological map of the municipalities of Casola Valsenio (**a**), Brisighella (**b**), Modigliana (**c**) and Predappio (**d**). FMA = Marnoso-Arenacea Formation, FAA = Blue Clays Formation.

*Casola Valsenio*

The municipality of Casola Valsenio is situated 30 km southwest of Imola (Fig. 1), covering an area of 84 km². It is predominantly known for its agricultural activities, including the cultivation of herbs, fruit orchards, and beekeeping. The underlying geology of the region is predominantly made up of the 'Unit 7' shows in Fig. 2a mainly composed by Marnoso-Arenacea Formation (FMA), a Tertiary flysch characterized by a finely interbedded sequence of marl and sandstone layers. This sequence was deposited in a deep marine foredeep basin during the Miocene epoch (Amy et al., 2006). The formation is characterized by a variable Arenites/Pelites (A/P) ratio, which reflects variations in



depositional environments within the basin. In Casola Valsenio, nearly the entire area features FMA
with an A/P ratio ranging from 1/3 to 3/1. The northern sector is distinguished by predominantly
sandstone-rice layers (A/P > 3) along with minor outcrops of Tectonized Clays (FPP) and Messinian
Gypsum (FGY) formations. Before May 2023, the Geological Survey had documented 730 ancient,
inactive landslides, primarily rock-block slides on cataclinal slopes that align with bedding planes.
Following the May 2023 disaster, this figure dramatically increased to 5572 new landslides (Fig. 2a).
Of these, debris slides (DS) and debris flows (DF) accounted for the vast majority, comprising 83%
(4,543 events) and 14% (789 events) of the total, respectively. Typically, these were initiated by
shallow failures of the thin soil layer (averaging 0.5-1 meters in depth) that overlies the Marnoso-
Arenacea bedrock (Fig. 3b-c-d). In line with Berti et al. (2025), DF events were further classified into
long-runout (DF1, 334 events) and limited-runout (DF2, 445 events), while DS events included 63
occurrences of limited-runout type (DS2, 1%).  Additionally, 17 earth landslides were recorded,
including earth flows (EF, 11 events; 0.2%) and earth slides (ES, 6 events; 0.1%). The area also
witnessed several large rock-block slides on cataclinal slopes (Fig. 3a). Although these accounted for
only 2% of the total landslides (110 events), their size was significantly greater than that of the debris
slides and flows, resulting in extensive damage. The May 2023 disaster notably disrupted local
infrastructure, blocking the main road and cutting off the southern region for weeks. Nearly 70 homes
and 200 roadways were severely affected.

*Brisighella*
The municipality of Brisighella borders Casola Valsenio to the east (Fig. 1) and covers 194 km². The
area is primarily composed of the 'Unit 7' composed by Marnoso-Arenacea Formation (FMA) (Fig.
2b). The northern part of the municipality, approximately 75 km² in size, consists mainly of the Blue
Clays Formation (FAA) (Fine-grain rocks 'Unit 1' in Fig. 1b), which is bordered by a narrow strip of
gypsum (Fig. 2b). Before the event, the Geological Survey had documented 2,100 landslides in the
area, mostly ancient and inactive rock-block and complex slides. Following the disaster, an additional
6342 landslides were recorded.
Of these, 61% were debris slides (DS) (3,890 events), while debris flows (DF) accounted for 16% of
the total (999 events), 6% of the with a long-runout (DF1) and 10% with limited-runout (DF2). Earth
slides (ES) and earth flows (EF), typically associated with badlands in clayey terrains, represented
22% of the total (1,428 events).. Lastly, rock-block slides accounted for 0.4% of the recorded events
(25 occurrences).. In the municipal territory, landslides blocked key access routes and damaged 66
homes and 261 roadways.

*Modigliana*
The municipality of Modigliana is located approximately 15 km to the east of Casola Valsenio (Fig.
1), and is one of the area most seriously affected by the event. The area spans 101 km² and is primarily
composed of the Marnoso-Arenacea Formation, FMA ('Unit 7') with an A/P ratio ranging from 1/3
to 3/1 (Fig. 2c). A small portion of Blue Clays, FAA (Unit 1) outcrops in the northeast. Before the
disaster, 795 landslides were documented, mostly complex and rock-block slides. After the May 2023
event, the number of landslides jumped to 6974, drastically reshaping the landscape. Of these, debris
slides were the most common, accounting for 77% of the total (5357 events). Debris flows (DF)
represented 19%, divided between long-runout (DF1, 573 events) and limited-runout (DF2, 733



events) types, while rock-block slides accounted for 1% of the total (69 events). Earth slides (ES) and
earth flows (EF) made up 3.5% (242 events). The disaster blocked the main access road for weeks,
affecting 51 homes and 155 roadways, and left the community to face long-term recovery challenges.

*Predappio*
The municipality of Predappio is situated around 30 km east of Casola Valsenio (Fig. 1) and spans an
area of 91 km². The geological framework of Predappio is mainly characterized by the Marnoso-
Arenacea Formation, FMA ('Unit 7') with an A/P ratio ranging from 1/3 to 3/1. The central part of
the municipality also features the Colombacci Formation ('Unit 8' in Fig. 1b), interspersed with
gypsum from the Gessoso-Solfifera Formation ('Unit 3'). Towards the northeast, the landscape
transitions to the Blue Clays formation ('Unit 1'), interbedded with deposits of the Colombacci
Formation and several occurrences of Pliocene sandstones (Borello Sandstones, 'Unit 7').
Before May 2023, 840 landslides had been recorded, primarily involving rock-block slides and debris
flows. Following the disaster, this number escalated to 6832. Of these, 80% were debris slides (5.476
events), while debris flows (DF) accounted for 11% (750) of the total including 419 long-runout
events (DF1) and 331 limited-runout events (DF2). Rock-block slides (RS), although only 0.5% of
the recorded events (31 occurrences), were notable for their significant size and potential impact.
Earth slides (ES) and earth flows (EF) represented 8% of the total (551 events).
The area experienced weeks of isolation due to several landslides blocking the main road, impacting
19 homes and damaging 138 roadways, again with long-lasting consequences.





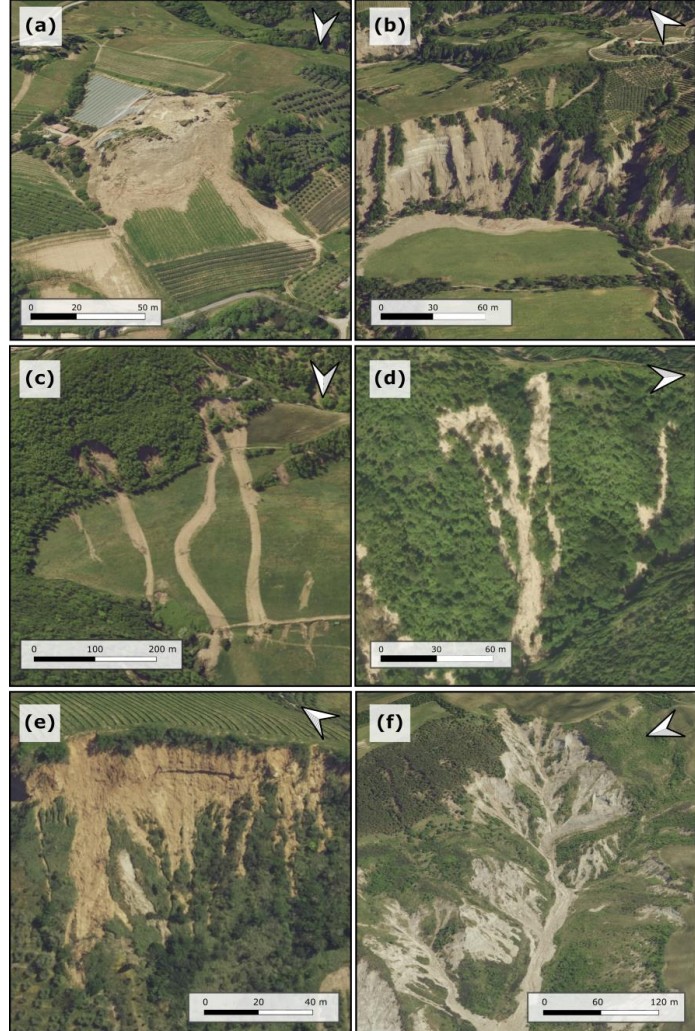

**Figure 3.** Examples of Landslides in Romagna from May 2023 event following Berti et al. 2025 classification.

**a**) Rockslide (RS) at Baccagnano, Brisighella, triggered during the night of May 2-3.
    (WGS84: 44.2069085, 11.7739792).

**b**) Debris slide (DS1) at Monte Albano, Casola Valsenio. (WGS84: 44.2356093, 11.6682287).

**c**) Debris flow with extended runout on gentle unforested slope (DF1) at Casola Valsenio.
    (WGS84: 44.20834, 11.64312).

**d**) Debris flow with limited runout on steep forested slope (DF2) at Modigliana.
    (WGS84: 44.15124, 11.75511).

**e**) Earth Slide (ES) at Brisighella. (WGS84: 44.23912, 11.78393).

**f**) Earth Flow (EF) at Predappio. (WGS84: 44.08947, 11.99627).

Images © Regione Emilia-Romagna, Geoportale 3D (https://mappe.regione.emilia-romagna.it)



**3. Methods**

3.1 Data

Deep-learning models for landslide mapping typically utilize a multi-layer input approach, integrating a combination of satellite or aerial imagery, topographical data, and land use information to capture the complex characteristics of landslides (Ghorbanzadeh et al., 2019, Meena et al., 2022). In our analysis, we considered the following seven layers (Fig. 4):

- L1) Post-event Sentinel-2 images: Four-bands (RGB+NIR) satellite images with a 10 m resolution, captured after the second rainfall event on May 23, 2023. They represent the first post-event Sentinel-2 images obtained with minimal cloud cover (Fig. 4b).

- L2) Pre-event Sentinel-2 images: Four-bands (RGB+NIR) satellite images with a 10 m resolution captured in May 2022, one year prior to the event. These images provide a similar vegetation state to that of May 2023, and were chosen to represent pre-event conditions due to the lack of cloud-free images in the two months preceding the event (Fig. 4a).

- L3) NDVI change Sentinel-2 map: Single-band raster generated by subtracting the pre-event NDVI map (Kriegler et al., 1969) derived from L2 from the post-event NDVI map created using L1 (Fig. 4c).

- L4) Post-event CGR images: Four bands (RGB+NIR) aerial photographs with a very high resolution of 0.2 m captured on May 23, 2023, shortly after the event. These images were committed by the Emilia-Romagna Region specifically for the disaster (Fig. 4f).

- L5) Pre-event AGEA images: Four bands (RGB+NIR) aerial photographs with a very high resolution of 0.2 m captured 3 years before the event (April to June 2020). These images were committed by the Agency for Agricultural Payments mainly for agricultural purposes (Fig. 4e).

- L6) NDVI change CGR map: Single-band raster generated by subtracting the pre-event NDVI map derived from L5 from the post-event NDVI map created using L4 (Fig. 4g).

- L7) Slope map: Single-band raster derived from the 5x5 m Digital Terrain Model (DTM) of the Emilia Romagna Region. The DTM, which is based on the 1:5000 scale Regional Technical Map, captures the pre-event slope morphology (Fig. 4d).

These layers were selected to simulate the diverse scenarios that might be encountered during an emergency, based on the available data. Table 2 details the potential combinations of the seven layers, with each combination corresponding to different scenarios explored in the study. In every case listed in Table 2, we assumed the availability of a slope map (L7), as numerous global DEMs such as SRTM (Farr et al., 2007; NASA SRTM, 2013), ASTER (NASA Earthdata Search, 2023), ALOS (Rosenqvist et al., 2007; ASF DAAC, 2023), and Copernicus DEM (ESA, 2019) are readily accessible for free.

Often, Sentinel-2 images (L1) might be the only data accessible following a large-scale disaster. The Sentinel-2 satellites, part of the European Copernicus program, revisit any point on Earth every 5 days and provide free 10 m resolution images in the visible and near-infrared spectrum, which are vital for various applications including disaster management (Wasowki et al., 2014; Yang et al., 2019; Ban et al., 2020). Numerous studies have utilized these open-access images to develop and evaluate automated methods for landslide mapping (Liu et al., 2021; Ghorbanzadeh et al., 2022; Lu et al.,



2023; Notti et al., 2023). Consequently, L1 can be considered the basic data layer available in the
aftermath of an event (case 1 in Table 2).
Frequently, though not always, it is possible to access pre-event Sentinel-2 images (L2) and compute
the NDVI change map (L3) to improve the identification of landslides triggered by an event.
However, these resources may not be available if the disaster was preceded by extended periods of
cloudy weather during which significant changes in vegetation state occurred. In our study, we
utilized cloud-free Sentinel-2 images taken during the same period the previous year, but this
approach isn't always feasible in regions with intensive agricultural activity, where the state of
vegetation can vary markedly from one year to the next. When pre-event Sentinel-2 images are
available, automated methods can be trained using L1, L2, and L3 (case 2).
In other cases, when financial resources permit, high-resolution images can be obtained soon after
the disaster. These images may come from satellite systems such as WorldView-3 and 4 (Maxar
Technologies, 0.31 m), GeoEye-1 (Maxar Technologies, 0.41 m), Pleiades 1A and 1B (Airbus, 0.5
m), IKONOS (Maxar Technologies, 0.82 m), or SkySat (Planet Labs, 0.9 m), or they can be captured
through dedicated aerial campaigns. The use of high-resolution imagery greatly enhances the
precision of landslide mapping as it allows for the detection of even the smallest features and the clear
identification of subtle geomorphic signs of instability. When high-resolution post-event images are
available, they can be used on their own (case 3), or in combination with Sentinel-2 images (case 4),
facilitating the capture of both small-scale and large-scale landslide features.
Less commonly, high-resolution images suitable for analysis are available before the event. The
utility of these images' hinges on their timing relative to the disaster. Ideally, pre-event high-
resolution images should capture the area immediately before the disaster to accurately pinpoint
landslides triggered by the event and minimize false positives resulting from agricultural or natural
vegetation changes. This requires appropriate satellite coverage of the area prior to the disaster. In
our case, high-resolution aerial images were obtained three years before the event (L5); however, the
general land cover remained unchanged, and no significant landslides had occurred in the area since
then. Automated methods can be applied by integrating both pre- and post-event high-resolution
images (case 5) or by also incorporating an NDVI change map derived from these datasets (case 6).
A final case considers the combination of all available layers L1-7 (case 7).
Although the post-event (L4) and pre-event (L5) aerial images, as well as the corresponding NDVI
change map (L6), were originally available at a very high spatial resolution of 0.2 m, their use at full
resolution posed major computational issues. Specifically, attempts to process these layers at native
resolution systematically resulted in out-of-memory (OOM) errors during model training and
inference. This was due to the large size of the input tensors generated from 0.2 m imagery over wide
areas, which exceeded the available GPU memory even when using optimized batch sizes and
patching strategies.
To overcome this limitation and ensure stable execution of the deep learning pipeline, we resampled
the L4, L5, and L6 layers to a coarser spatial resolution of 2 m. This down sampling allowed us to
retain sufficient spatial detail for landslide identification, while significantly reducing the memory
footprint of the input data. While the original 0.2 m resolution would have provided extremely fine
detail, it exceeded the requirements and computational feasibility for a regional-scale deep learning
analysis. In contrast, the 2 m resolution still captures the essential geomorphological features relevant





to landslide detection and represents a practical and effective compromise. This trade-off is consistent
with the flexibility of deep-learning models, which can learn robust spatial representations from
moderate-resolution inputs when properly trained and validated.

**Table 2**. Overview of the various input layer configurations used during the training processes. 'NDVI'
refers to the 'ΔNDVI-CGR' Change map created using CGR imagery, while 'ΔNDVI-S2' refers to
the NDVI Change map created using Sentinel-2 imagery.

| Name | Model | Sentinel2 post [L1] | Sentinel2 pre [L2] | Sentinel2 ΔNDVI [L3] | CGR post [L4] | AGEA pre [L5] | CGR ΔNDVI [L6] | Slope [L7] |
|------|-------|------|------|------|------|------|------|------|
| U1 | U-Net | ✓ | X | X | X | X | X | ✓ |
| S1 | SegForm | ✓ | X | X | X | X | X | ✓ |
| U2 | U-Net | ✓ | ✓ | ✓ | X | X | X | ✓ |
| S2 | SegForm | ✓ | ✓ | ✓ | X | X | X | ✓ |
| U3 | U-Net | X | X | X | ✓ | X | X | ✓ |
| S3 | SegForm | X | X | X | ✓ | X | X | ✓ |
| U4 | U-Net | ✓ | ✓ | ✓ | ✓ | X | X | ✓ |
| S4 | SegForm | ✓ | ✓ | ✓ | ✓ | X | X | ✓ |
| U5 | U-Net | X | X | X | ✓ | ✓ | X | ✓ |
| S5 | SegForm | X | X | X | ✓ | ✓ | X | ✓ |
| U6 | U-Net | X | X | X | ✓ | ✓ | ✓ | ✓ |
| S6 | SegForm | X | X | X | ✓ | ✓ | ✓ | ✓ |
| U7 | U-Net | ✓ | ✓ | ✓ | ✓ | ✓ | ✓ | ✓ |
| S7 | SegForm | ✓ | ✓ | ✓ | ✓ | ✓ | ✓ | ✓ |





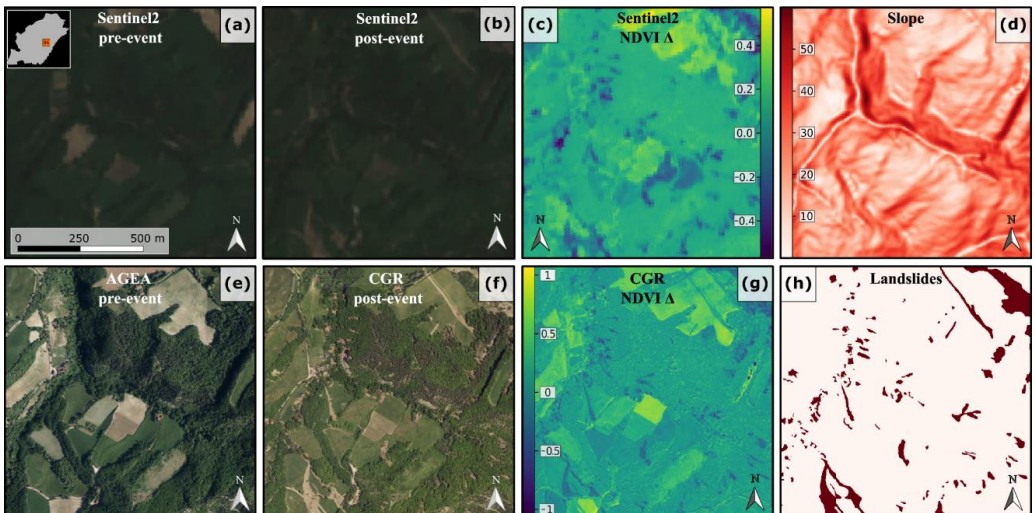


**Figure 4.** Example of the layers included in case 7, Tile No. 86. (**a**) Pre-event Sentinel-2 image
(2022); (**b**) Post-event Sentinel-2 image; (**c**) NDVI change map derived from Sentinel-2 imagery; (**d**)
Slope map; (**e**) AGEA imagery (2020), pre-event; (**f**) CGR aerial image (2023), post-event; (**g**) NDVI
change map derived from high resolution imagery; (**h**) Binary raster showing manually mapped
landslides (ground truth).

3.4 Deep Learning Semantic Segmentation Models
*U-Net*
U-Net is a convolutional neural network architecture originally designed for biomedical image
segmentation by Ronneberger et al. (2015). It features an encoder-decoder structure where the
encoder progressively reduces the spatial dimensions through convolutional and max-pooling layers,
capturing contextual and high-level features. The decoder then up samples the features using
transposed convolutions and concatenates them with corresponding encoder features through skip
connections, which help retain spatial information and improve localization accuracy. This
combination of context and localization has proven highly effective for segmentation tasks across
various domains. U-Net's effectiveness in landslide mapping has been demonstrated in recent studies
(Meena et al. 2021, 2022; Ghorbanzadeh et al. 2022, 2023; Nava et al. 2022).
In our study, we implemented the U-Net architecture manually in Python. The input shape of the
images was defined according to the specific dimensions of the data used, ensuring the model could
effectively process the input imagery. We set the number of scales to 2, which determines the number
of downsampling steps in the encoder, and each scale included 2 convolutional layers, which helped
capture intricate details and patterns within the data. The initial number of convolutional filters was
set to 64. This parameter defines the number of filters applied in the first convolutional layer, which
influences the model's ability to learn from the input data. As the network progresses through the
layers, the number of filters increases, allowing the model to capture more complex features.
Our model was designed to output 2 classes: landslide and non-landslide. This binary classification
approach enabled us to effectively differentiate between affected and unaffected areas in the imagery.



The model was compiled using the Adam optimizer (Kingma and Ba, 2017), a learning rate of 0.0001
and a Dice loss function for 300 epochs. The early stopping mechanism was introduced with a
patience of 20 epochs to prevent overfitting, based on validation loss monitoring. All analyses were
conducted using Python 3.8.18, TensorFlow 2.5.0, and CUDA 11.2.
*SegFormer*
SegFormer is a state-of-the-art architecture for semantic segmentation of multispectral landslide
imagery (Xie et al., 2021). SegFormer, is designed to overcome limitations of previous vision
Transformer models, such as their high computational cost, reliance on fixed input resolutions due to
positional encodings, and inefficient multi-scale feature aggregation (Dosovitskiy et al., 2020). While
the original SegFormer architecture was designed for 3-channel RGB images, the
SegformerForSemanticSegmentation implementation from Hugging Face Transformers library
(Wolf et al., 2020) allows for flexible input channel configuration. We leveraged this feature to adapt
SegFormer to our multi channels imagery.
The SegFormer architecture consists of two main components (Xie et al., 2021):
1.  Hierarchical Transformer Encoder: This encoder employs a novel hierarchical structure with
progressively larger stride and patch sizes, allowing the model to capture both fine and coarse
features efficiently. It uses an efficient self-attention mechanism without positional encoding
and incorporates a Mix-FFN block that includes a depth-wise convolution to enhance local
spatial information aggregation.
2.  Lightweight All-MLP Decoder: Unlike complex decoder designs in previous models,
SegFormer uses a simple multi-layer perceptron (MLP) decoder. This decoder fuses multi-
level features from the encoder through a unified set of MLP layers, achieving per-pixel
prediction without the need for complex operations like FPN or dilated convolutions.
SegFormer's design eliminates the need for positional encoding, allowing it to handle variable input
resolutions without interpolation. This feature is particularly beneficial for our landslide imagery,
which can vary in resolution and scale.
Our model was configured as the variant MiT-b0 (Xie et al., 2021) with the following parameters:
▪  Hidden sizes: [32, 64, 160, 256]
▪  Encoder depths: [2, 2, 2, 2]
▪  Decoder hidden size: 256
The model was trained using Cross-Entropy Loss as the criterion. We employed the Adam optimizer
with a learning rate of 0.001 for 300 epochs. The early stopping mechanism was introduced with a
patience of 20 epochs to prevent overfitting, based on validation loss monitoring. All analyses were
conducted using Python 3.8.10, PyTorch (Paszke et al., 2019) 1.9.0, and CUDA 11.1.

3.5 Training-Testing Split
While conventional practice often dictates a 70/30 or 80/20 ratio for training and testing datasets
(Hastie et al., 2009), we adopted a different approach to simulate real-world emergency mapping
conditions. The model was exclusively trained on data from the Casola Valsenio municipality and



subsequently applied to three additional municipalities, Predappio, Modigliana, and Brisighella, for
independent testing. This setup simulates a realistic emergency response scenario where; after
mapping a relatively small area, the model is deployed to rapidly map other affected areas, relying
solely on prior training.
To facilitate analysis, the train area was segmented into 1x1 km grid tiles, as illustrated in Fig. 5. Each
1 km² tile corresponds to 512x512 pixels in the CGR imagery when resampled at 2m resolution.
Generally, a single grid tile in these areas encompasses over 50 landslides, providing a robust
representation of landslide patterns.
For model training, we adopted a stratified random sampling approach. Out of the 97 total tiles in
Casola Valsenio, 62% (60 tiles) were randomly allocated for training, 15% (15 tiles) for validation,
and the remaining 23% (22 tiles) were reserved for initial testing and further validation (Fig. 5). This
division was designed to maximize the training set while ensuring sufficiently large and reliable
validation and test sets, given the need to apply the model to other municipalities.

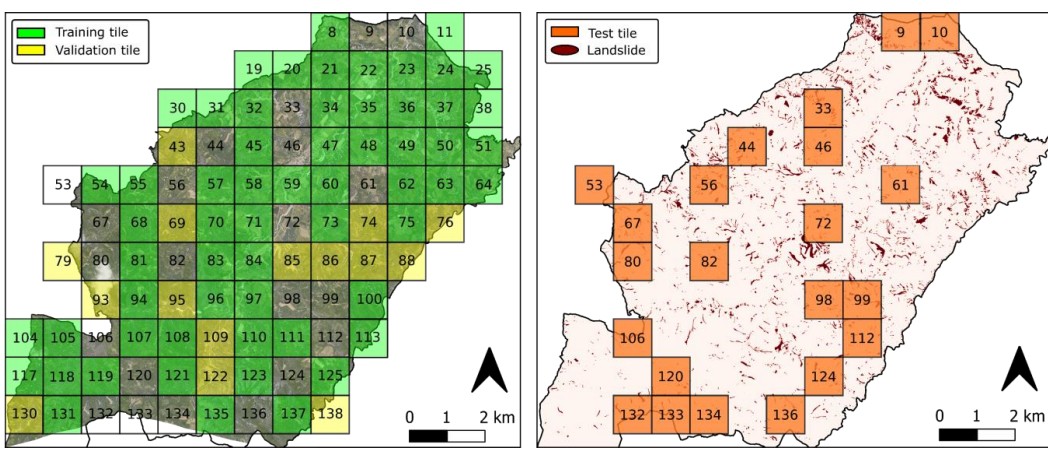


**Figure 5.** Training and test tiles used for models' construction in Casola Valsenio, illustrating the
spatial split of the area.

3.6 Model Application
Following the training phase, we applied the models to the municipalities of Predappio, Modigliana,
and Brisighella. Crucially, while ground truth (Fig. 1a) data was available for these municipalities,
we deliberately chose not to use it for model training. Instead, this data was reserved exclusively for
evaluating the performance of our models.
A significant constraint of the analysis arises from the predominance of the Marnoso-Arenacea
Formation (FMA) in the training dataset from Casola Valsenio (Fig. 3a). To assess the models' ability
to map landslides in varied geological contexts, we divided the Brisighella municipality into two
distinct geological zones: one dominated by the Marnoso-Arenacea Formation (Brisighella FMA)
and another characterized by the predominance of fine-grained rock from the Pliocene Blue Clays
(Brisighella FAA) (Fig. 3d). This latter area is characterized by earth flows (EF) and earth slides (ES),
which exhibit fundamentally different morphological, and soil cover features compared to landslides



present in the Marnoso-Arenacea Formation. Figure 6 shows representative examples of these
contrasting landslide types. The FAA zone, underlain by the Pliocene Blue Clays, is typified by slow-
moving earth flows (EF) and earth slides (ES), which often occur within badland-like morphologies.
These landslides commonly exhibit an elongated, teardrop-like form with a lobate toe, shaped by
multiple episodes of reactivation over time.The vegetative cover is often sparse or degraded, as
typically observed in clay-rich soils.
In contrast, landslides in the FMA zone, both in Brisighella and in the training area of Casola
Valsenio, are dominated by debris slides (DS) and debris flows (DF). These failures typically
originate from thin, unconsolidated colluvial layers that overlie the flysch bedrock, and are triggered
by intense rainfall infiltration. The resulting movements are generally sudden, with well-defined scarp
and runout zones, and affect areas with denser vegetation cover.
By explicitly separating these two lithological domains within Brisighella, we aimed to evaluate
whether a model trained solely on FMA-type landslides could generalize to areas with fundamentally
different geological and geomorphological settings. This setup also allowed us to assess the
sensitivity of the model to lithology-driven differences in landslide type, geometry, and spatial
pattern—an essential step toward developing mapping tools that are robust across variable terrain
conditions.

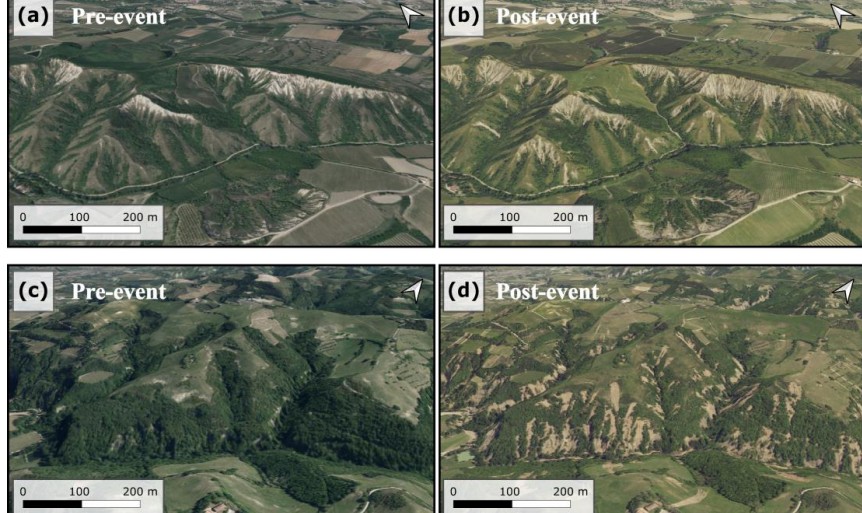

**Figure 6.** Comparison examples are shown between pre-event (**a**) and post-event (**b**) earth flow (EF)
landslides in the Blue Clays Formation (FAA) and pre-event (**c**) and post-event (**d**) debris slide (DS)
in the Marnoso-Arenacea Formation (FMA), both triggered by the May 2023 disaster in Brisighella.
Images © Regione Emilia-Romagna, Geoportale 3D (https://mappe.regione.emilia-romagna.it)

The final evaluation of model generalization was conducted on the entire set of grid tiles from the
three remaining municipalities: 90 tiles from Predappio, 105 from Modigliana, and 196 from
Brisighella. As such, although the core training-validation-testing split within Casola Valsenio





followed a 62/15/23 ratio, the effective test set includes not only the 22 held-out tiles from Casola
Valsenio, but also 100% of the 413 tiles from the other three municipalities, a spatial hold-out test
designed to assess cross-municipality generalization.

**Table 3.** Overview of the utilization of 1 km² tiles across the analyzed municipalities, illustrating how
each area was segmented for model training, validation, testing and prediction purposes.

| Area | Tiles [1 km²] | Use |
|---|---|---|
| Casola Valsenio | 97 | 60 train / 15 validation/ 22 test |
| Brisighella FMA | 116 | Prediction |
| Brisighella FAA | 80 | Prediction |
| Modigliana | 105 | Prediction |
| Predappio | 90 | Prediction |



3.7 Evaluation Metrics and Expert Judgment
*Metrics*
To assess the performance of our landslide detection models, we employed two widely used metrics
in semantic segmentation tasks: the F1 score and the Intersection over Union (IoU). These metrics
were calculated on the entirety of the test images, rather than averaging scores across individual cells,
to provide a comprehensive evaluation of the models' performance.
The F1 score, also known as the Dice coefficient, is the harmonic mean of precision and recall
(Tharwat, 2020). It is particularly useful in imbalanced classification problems, such as landslide
detection, where the positive class (landslide) is typically much smaller than the negative class (non-
landslide). The F1 score is calculated as:
$$\text{F1} = 2 \cdot \frac{Precision \cdot Recall}{Precision + Recall}$$  (1)
where Precision = TP / (TP + FP) and Recall = TP / (TP + FN), with TP, FP, and FN representing
True Positives, False Positives, and False Negatives, respectively.
The Intersection over Union (IoU), also known as the Jaccard index, is another common metric for
assessing segmentation accuracy (Rezatofighi et al., 2019). It measures the overlap between the
predicted segmentation mask and the ground truth mask. The IoU is calculated as:
$$\text{IoU} = \frac{TP}{TP + FP + FN}$$  (2)
Both metrics range from 0 to 1, with 1 indicating perfect prediction. By using these metrics on the
entire test images, we can evaluate how well our models perform in detecting and delineating
landslides across varied landscapes and geological contexts.

*Expert Judgment*
In automated landslide mapping, a higher algorithmic score does not necessarily correspond to a
higher quality map (Zhang et al., 2018; Isensee et al., 2021).  This issue is particularly relevant in





automated landslide mapping, where the practical utility of the map often depends more on its interpretability and coherence with geomorphological reality than on its statistical performance alone.

For example, a model might overpredict large landslide areas to optimize pixel-wise metrics like IoU, while neglecting smaller or fragmented features that are critical for field validation and emergency response. Conversely, models may introduce noise or irregular boundaries that artificially increase recall but reduce the practical readability of the map.

While this issue is typically addressed by selecting evaluation metrics that emphasize specific components of the confusion matrix (e.g., recall over precision), in emergency scenarios not all components carry the same operational weight. For instance, in crisis management, a higher recall may be preferred to avoid missing active landslides, even if this comes at the cost of more false positives. In contrast, in densely urbanized areas or during resource-constrained interventions, precision may become more important to avoid unnecessary alarms and misallocation of efforts.

To address this challenge, three of the authors in this study provided their expert judgment on the quality of the automatically generated landslide maps. This evaluation was conducted without prior knowledge of the scores obtained by these maps or the training input layers used to generate them.

The assessment focused on seven common error types frequently encountered in automated landslide mapping:

E1 – False positives in agricultural fields, where cultivated land is mistakenly classified as landslide;

E2 – False positives along roads and in urban areas, where the absence of vegetation may be misinterpreted as landslide activity;

E3 – False positives along riverbeds, often due to confusion with recent sediment scouring or bedload deposits;

E4 – False negatives in areas with known landslides;

E5 – Over segmentation, referring to the fragmentation of a single landslide into multiple small polygons;

E6 – Inaccurate delineation of landslide boundaries;

E7 – Omission errors in shadowed areas, such as steep, shaded slopes or regions obscured by cloud cover.

For each error type, the experts assigned a score of 0 (limited error), 1 (moderate error), or 2 (relevant error) to indicate the severity of the issue. These scores were then summed and normalized to calculate an Expert-based Performance Index (EPI):

$$EPI = \frac{14 - \sum_{i=1}^{i=7} E_i}{14}$$

The index ranges from 0 to 1, with 1 representing the highest mapping performance. This evaluation was independently carried out by each of the three experts for all 16 cases included in the analysis (see Table 2). The experts assessed the model cases without prior knowledge of the specific configurations, thereby minimizing potential bias.






**4. Results**

4.1 Model results and performance

The results of the analysis are presented in Table 4. The table details the F1 and IoU scores obtained
by comparing the automated landslide maps produced by two machine learning models (U=U-Net;
S=SegFormer) against the manually created maps. The rows list the seven different combinations of
input layers, numbered from 1 to 7, to represent a progressively richer dataset (refer to Table 1). The
models were initially trained using data from Casola Valsenio and subsequently applied to four
additional areas (Predappio, Modigliana, Brisighella FMA, and Brisighella FAA). Overall, this
combination of 7 input datasets, 4 areas, and 2 models resulted in 56 automated landslide maps.
In the tested configurations, the models U6, U7, and S7 achieved the highest F1 and IoU scores. These
models incorporate nearly all (U6) or all (U7-S7) of the seven input layers, and clearly benefit from
the high-resolution pre- and post-event images. However, the performance gains from utilizing all
available layers are relatively modest. Notably, model S2, which employs only Sentinel-2 images and
Sentinel-2 NDVI, achieves results close to those of the top-performing models, with differences in
F1 and IoU scores frequently around a margin of ~0.01. Similar observations apply to models U4 and
S4, which rely on Sentinel-2 products and post-event high-resolution images. This suggests that while
increasing input information enhances model performance, even more streamlined configurations can
still yield robust and competitive mapping outcomes. The good performance of U4 and S4 models
highlights the effectiveness of integrating high-resolution imagery alongside Sentinel-2-derived
indices, even when fewer spectral inputs are available.



















**Table 4.** Quality of the automatic mapping, based on F1 and IoU scores, achieved by U-Net and SegFormer algorithms with different combinations of input layers, detailed in Table 2.

| Models' Name | Casola Valsenio | | Predappio | | Modigliana | | Brisighella FMA | | Brisighella FAA | |
|---|---|---|---|---|---|---|---|---|---|---|
| | F1 | IoU | F1 | IoU | F1 | IoU | F1 | IoU | F1 | IoU |
| U1 | 0.56 | 0.39 | 0.42 | 0.27 | 0.48 | 0.31 | 0.50 | 0.33 | 0.32 | 0.19 |
| S1 | 0.47 | 0.31 | 0.46 | 0.30 | 0.45 | 0.29 | 0.46 | 0.30 | 0.34 | 0.21 |
| U2 | 0.64 | 0.47 | 0.52 | 0.35 | 0.55 | 0.38 | 0.57 | 0.40 | 0.38 | 0.24 |
| S2 | 0.62 | 0.45 | 0.54 | 0.37 | 0.57 | 0.40 | 0.57 | 0.40 | 0.47 | 0.31 |
| U3 | 0.70 | 0.54 | 0.39 | 0.24 | 0.45 | 0.29 | 0.55 | 0.38 | 0.29 | 0.17 |
| S3 | 0.54 | 0.37 | 0.44 | 0.28 | 0.44 | 0.29 | 0.49 | 0.32 | 0.32 | 0.19 |
| U4 | 0.72 | 0.56 | 0.55 | 0.38 | 0.58 | 0.41 | 0.62 | 0.45 | 0.47 | 0.31 |
| S4 | 0.68 | 0.52 | 0.59 | 0.42 | 0.60 | 0.43 | 0.62 | 0.45 | 0.49 | 0.33 |
| U5 | 0.71 | 0.55 | 0.53 | 0.36 | 0.56 | 0.39 | 0.62 | 0.45 | 0.50 | 0.34 |
| S5 | 0.65 | 0.48 | 0.56 | 0.39 | 0.56 | 0.39 | 0.60 | 0.43 | 0.43 | 0.28 |
| U6 | 0.72 | 0.56 | 0.56 | 0.38 | 0.59 | 0.42 | 0.63 | 0.46 | 0.53 | 0.36 |
| S6 | 0.65 | 0.48 | 0.53 | 0.36 | 0.54 | 0.37 | 0.58 | 0.41 | 0.41 | 0.26 |
| U7 | 0.73 | 0.57 | 0.56 | 0.39 | 0.60 | 0.42 | 0.63 | 0.46 | 0.53 | 0.36 |
| S7 | 0.66 | 0.49 | 0.60 | 0.43 | 0.60 | 0.43 | 0.61 | 0.44 | 0.52 | 0.35 |

Figure 7 shows the F1-score trends across all seven input layer combinations (cases 1 to 7) for the five evaluated areas. In both models, a marked increase in performance is observed between case 1 and case 2, particularly in Casola Valsenio, Modigliana, and Brisighella FAA. This highlights the importance of including NDVI change information alongside Sentinel-2 imagery to reach better segmentation performance. From case 2 onward, the models tend to stabilize, with U-Net showing a more pronounced benefit from additional input layers, especially in cases 4 to 7. SegFormer, by contrast, reaches near-plateau performance earlier and exhibits smaller gains as layers are added.

Performance across the different municipalities reveals further differences between the two models. U-Net shows a clear separation in F1-scores depending on the area: it performs best in Casola (reaching 0.73 in case 7), moderately well in Brisighella FMA, Modigliana, and Predappio, and significantly worse in Brisighella FAA (with 0.29 in case 3 and 0.53 in case 7). This trend reflects U-Net's sensitivity to geological conditions.

SegFormer, on the other hand, provides more uniform scores across areas, although generally lower than U-Net in Casola. Its performance ranges between 0.47–0.68 in Casola and remains around 0.55–0.60 in Predappio, Modigliana, and Brisighella FMA. Brisighella FAA again represents a challenge,



with F1-scores fluctuating and even slightly declining from case 2 onward, suggesting that additional
input layers do not consistently improve results in this more complex setting.
Overall, both models identify Brisighella FAA as the most difficult area, with the lowest F1-scores
recorded. However, while U-Net appears to benefit more from richer input combinations, SegFormer
demonstrates greater generalization ability across areas with different lithological and morphological
characteristics.

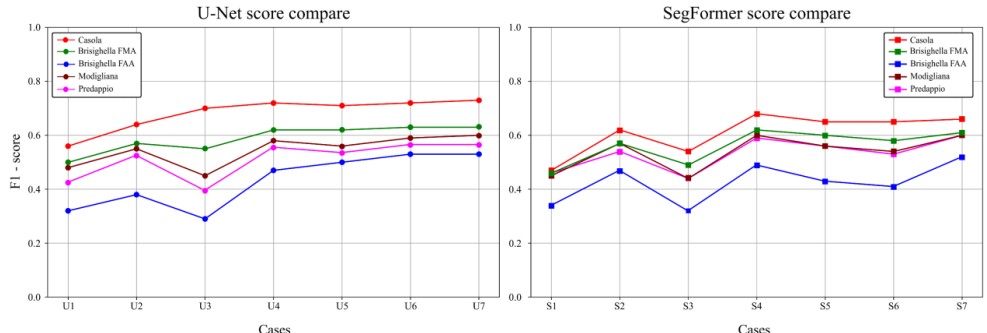


**Figure 7.** F1-score comparison across all seven input layer combinations (cases 1–7) for each
municipality, as obtained by the U-Net and SegFormer models.
As a complement to the quantitative evaluation, Figures 8 and 9 provide illustrative examples of
model performance in real-world conditions. These visual comparisons, focusing on the best-
performing configurations (U-Net case 7 and SegFormer case 7), highlight both successful detections
and common failure modes observed across the test municipalities.
Figure 8 displays cases where both models performed well, accurately mapping landslides in areas
with widespread vegetation removal, typical of debris slides (DS) and debris flows (DF). These
landslide types, which together accounted for about 81% of all events in May 2023, were consistently
recognized, especially in severely impacted zones where spectral and texture changes were
prominent. Figure 9, by contrast, presents a selection of cases where both models encountered notable
difficulties, leading to different types of misclassifications:
▪ Casola Valsenio: Despite being the training area, the models occasionally misinterpreted
the spatial extent of landslide polygons. In some cases, large landslides are fragmented
into several smaller segments, resulting in False Negatives and underestimation of the
actual affected area. This issue is likely due to the complex shape and internal
heterogeneity of the landslide footprints.
▪ Modigliana: Both U7 and S7 show a tendency to produce False Positives in cultivated
fields. These areas are subject to seasonal changes due to agricultural activities such as
plowing and vegetation regrowth, which often produce surface variations that resemble
the spectral signatures of recent landslides. In particular, disturbed or freshly tilled soil
patches in post-event imagery may be incorrectly classified as landslides.
▪ Brisighella FMA: In this area, False Positives were frequently mapped along riverbanks,
likely caused by localized color differences between the pre- and post-event images. These



differences may be associated with flood-related sediment deposition or changes in water level and channel morphology following localized inundations. Additionally, some False Positives were recorded on recently constructed buildings, especially where bare roofs or fresh concrete surfaces produced strong spectral contrasts. These elements, not present in the pre-event imagery, appear as "new" disturbed areas, misleading the models.

- Brisighella FAA: This area poses the most significant challenge for both models. Dominated by badlands developed in Blue Clay formations—geologically and morphologically distinct from the Marnoso-Arenacea Formation of the training area— Brisighella FAA yielded high rates of False Positives. The stark lithological difference appears to have limited the models' ability to generalize to this terrain, underlining the importance of including lithological variability in training datasets.

- Predappio: In this area, errors are primarily due to landslides occurring in shadowed regions of the post-event imagery. The lower contrast and altered color profiles in these zones reduced the models' ability to correctly detect landslide features, leading to partial detection or complete omission (False Negatives). These results suggest that terrain shading, common in steep valleys, remains a limitation for optical-image-based mapping.

Together, these examples illustrate both the strengths and the current limitations of the models when applied in real-world scenarios. While debris-type landslides in well-illuminated, vegetated settings were reliably mapped, generalization to areas with different lithologies, land use, or illumination conditions remains an area for improvement.

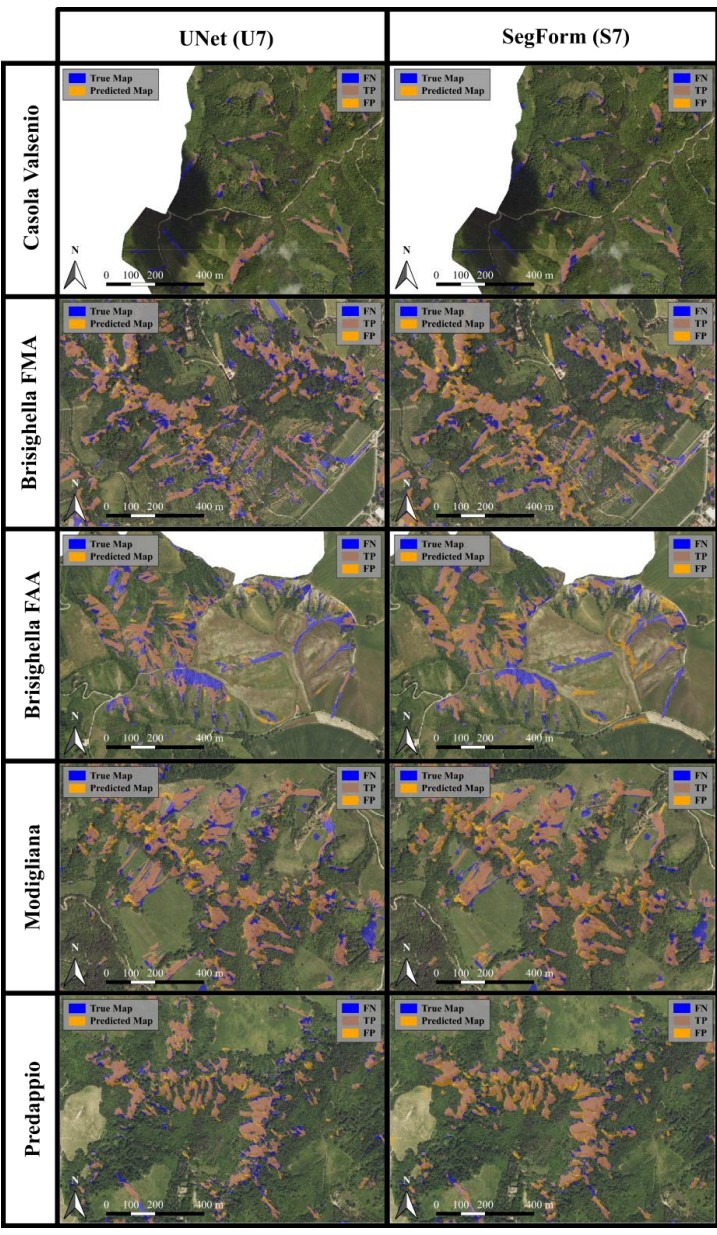

**Figure 8.** Comparative analysis of landslide mapping results using U7 and S7 models across different municipalities. The figure illustrates the spatial distribution and extent of landslides as identified by the U7 (UNet-based) and S7 (SegFormer-based) models in Casola Valsenio, Predappio, Modigliana, Brisighella FMA, and Brisighella FAA. Both models were trained using pre- and post-event imagery, slope data, and NDVI from CGR. The comparison highlights the similarities and differences in landslide detection capabilities between the two models, particularly in areas dominated by Debris Slides (DS).

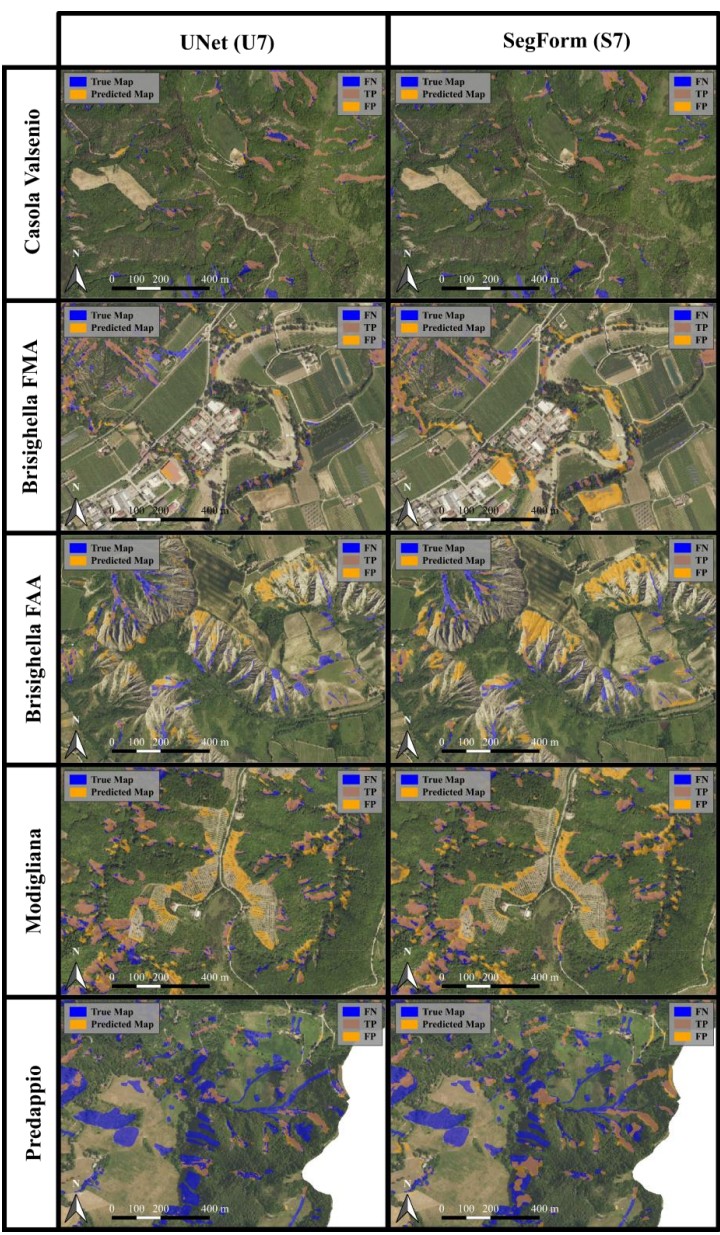

**Figure 9.** Representative examples of challenging scenarios for landslide detection using U7 and S7 models across municipalities. This figure illustrates common situations where both models encounter difficulties in accurately mapping landslides, highlighting the limitations and potential areas for improvement in the machine learning approach. The examples are drawn from Casola Valsenio, Predappio, Modigliana, Brisighella FMA, and Brisighella FAA, showcasing the diverse geological and environmental conditions that can impact model performance.



4.2 Expert judgement
Figure 10 presents a comparison between the F1-scores of all model cases and the EPI (Expert
Performance Index) values derived from expert assessments (see Section 3.7). The EPI values shown
in the charts reflect evaluations from three experts, who assessed the automated landslide maps
produced by the two models and assigned weights to seven common detection errors.
In general, the F1-scores align well with the EPI values obtained from expert judgement. Although
the F1-scores exhibit limited variability, the trend of improved performance with increasing model
complexity is also shown by the expert-based evaluations. Notably, several performance fluctuations
are similarly interpreted: model U3 is clearly recognized as underperforming, while model U7 is
consistently considered one of the best by both quantitative scores and expert opinion.
Some notable discrepancies emerge for certain models that were rated as the best by a single expert
but not by F1-scores or the other experts. For instance, model S5 received the highest score from
Expert 1, model U2 from Expert 2, and model U4 from Expert 3, despite not being ranked similarly
by others. These differences reflect the specific areas each expert emphasized during evaluation, as
well as the inherent subjectivity of expert judgment. However, the trend of the mean EPI (dashed
lines in Fig.10), which helps reduce individual biases, closely follows that of the F1-score.

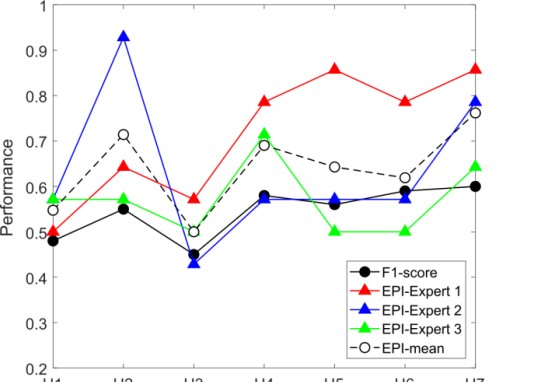 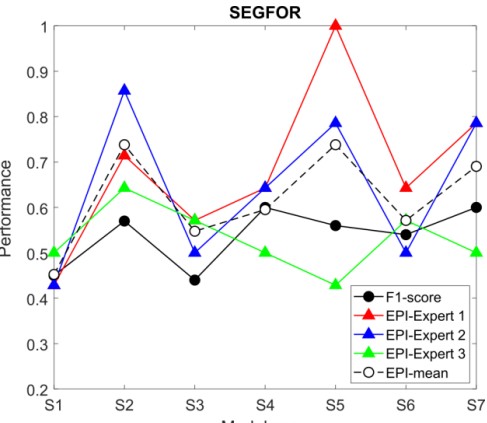


**Figure 10**. Comparison of model performance based on F1-scores and expert-based evaluations from
three independent experts.

**5. Discussion**
One of the key observations from this study concerns the differing sensitivity of U-Net and
SegFormer to input data combinations, particularly in the context of emergency response mapping.
U-Net's performance varies significantly depending on the layers used, while SegFormer yields more
consistent results across different configurations. This contrast is likely rooted in their architectural
differences. U-Net, designed for pixel-wise segmentation, relies heavily on the spatial and spectral
characteristics of the input data, making it more sensitive to the quality and availability of
supplementary layers such as pre-event imagery or NDVI change maps. SegFormer, being a



transformer-based model, is better equipped to capture global context, thus reducing dependency on
specific input combinations and improving robustness under complex geological conditions.
This architectural contrast becomes even more evident when models are transferred across different
geological domains. As shown in the results (Figure 7), U-Net exhibits a marked decline in
performance when moving from areas geologically similar to the training site (e.g., Predappio,
Modigliana) to Brisighella FAA, where lithological conditions differ significantly. This suggests that
U-Net has limited generalization capacity beyond the Marnoso-Arenacea Formation (FMA), and
struggles in fine-grained, clay-rich terrains like those found in the FAA domain. SegFormer, although
generally more stable, also underperforms in Brisighella FAA, highlighting the broader challenge of
transferring models across heterogeneous geological settings.
To further investigate the role of geology in model generalization, an additional analysis was
conducted to assess the impact of including lithology among the input layers. For instance, comparing
Case S3 (RGB + slope) with the same configuration plus a lithology layer did not yield significant
improvements in Casola, Predappio, Modigliana, or Brisighella MA. These areas are predominantly
characterized by the Marnoso-Arenacea Formation ("Unit 7"), which is already well represented in
the training dataset (Figure 2a), meaning the lithology layer provides redundant information.
However, when the same model was applied to Brisighella FAA, where lithology is dominated by
Blue Clays ("Unit 1"), it failed to detect any landslides, resulting in a very low F1-score of 0.02. This
highlights a key limitation: simply adding lithology as a static input layer is insufficient for ensuring
generalization. If the training dataset is not lithologically balanced, the model may reinforce existing
biases, associating landslide occurrence primarily with "Unit 7" and failing to detect events in "Unit
1".
This finding underscores a fundamental point: effectively incorporating lithological information into
AI-based landslide mapping requires more than providing it as an input feature. The training dataset
itself must include landslide polygons from a representative range of lithological settings. When the
model was explicitly trained and tested on Brisighella FAA using the same configuration (Case S3:
RGB + slope), the F1-score increased from 0.32 to 0.41, confirming that geological consistency in
the training data substantially improves the model's ability to detect landslides in previously
underrepresented domains.
Although the F1-scores achieved in this study may appear modest compared to some machine
learning benchmarks (https://huggingface.co/papers/trending), they represent a substantial
accomplishment in the context of landslide mapping. Unlike conventional image classification tasks,
landslide mapping requires the precise delineation of complex and irregular shapes across highly
variable terrain. Our models aim to produce a realistic representation of landslide extents, including
subtle transitions and intricate morphologies, rather than simplified or generalized outputs. This level
of accuracy is difficult to attain, yet essential for practical applications in emergency response and
hazard assessment.
In this regard, the potential of these automated approaches in emergency scenarios could be
considerable. The ability to rapidly generate landslide maps of comparable quality to expert-drawn
products could greatly accelerate initial response efforts. By reducing the need for time-consuming
manual digitization, these methods could save weeks of work, an especially critical advantage during
crisis events (Berti et al. 2025). The automatically generated maps may serve as a robust initial



product, allowing practitioners to focus on refinement and validation rather than starting from scratch, thereby enabling the delivery of a high-quality final map in a fraction of the time required for full manual mapping.

A further consideration is the use of automated landslide maps for identifying damage and at-risk structures. Following the May 2023 disaster, the Emergency Commission determined that all buildings located within landslide boundaries or within a 20-metre buffer were to be considered at risk and potentially subject to relocation. To evaluate the usefulness of our automated mapping products for this task, we compared the buildings identified automatically with those derived from manual mapping. Figure 11 presents the confusion matrix for the 16 modelled cases. The F1-scores indicate that "U2", "U4", "S6", and "U7" are the most accurate CNN outputs, successfully identifying on average 528 out of 654 buildings at risk (~0.78 F1-score). These results demonstrate the models' ability to estimate the spatial extent of the phenomenon, even when using Sentinel-2 imagery at 10 m resolution (U2). Nevertheless, all models produced both false positives (incorrectly flagged buildings) and false negatives (overlooked buildings). Consequently, manual verification remains essential when high accuracy is required, and misclassifications cannot be tolerated.

Even a small number of misclassified structures may have serious consequences when inhabited buildings are involved. This raises a key question: how can model performance be further improved to support more effective and precise emergency response efforts in the future? Several avenues could be explored:

1. Higher-resolution input data: As discussed in Section 3.1, high-resolution aerial imagery (20 cm/pixel) and NDVI change maps were down sampled to 2 m/pixel to reduce computational load. Using the original resolution, while computationally expensive, could yield improved detail and model performance (Wang et al., 2022).

2. Improved tiling strategies: The current approach relies on fixed 496×496 px tiles for training, validation, and testing (Section 3.5). This can split landslide polygons at tile boundaries and reduce spatial context. As noted by Abrahams et al. (2024), employing sliding windows or full-scene processing could mitigate these issues and enhance segmentation quality.

3. Data augmentation: No augmentation techniques were used in this study. Simple operations such as flipping, rotation, or brightness adjustment, shown by Prakash et al. (2021) could improve CNN robustness reducing overfitting and improve generalizability.

Finally, although current approaches enable rapid mapping of landslides, their ability to assess structural exposure remains limited. Buildings located just outside mapped polygons may still be at significant risk, which a uniform 20 m buffer may fail to capture adequately. A more advanced solution is proposed by BGC Engineering (2010), who incorporate topographic features and estimated landslide runout directions to dynamically assess exposure. In this framework, risk is evaluated based on terrain morphology, flow paths, and downslope connectivity, offering a more realistic representation of potential impacts compared to distance-based buffers. Such models could represent a major step forward in automated damage assessment, complementing segmentation outputs with geomorphologically informed prioritization strategies.



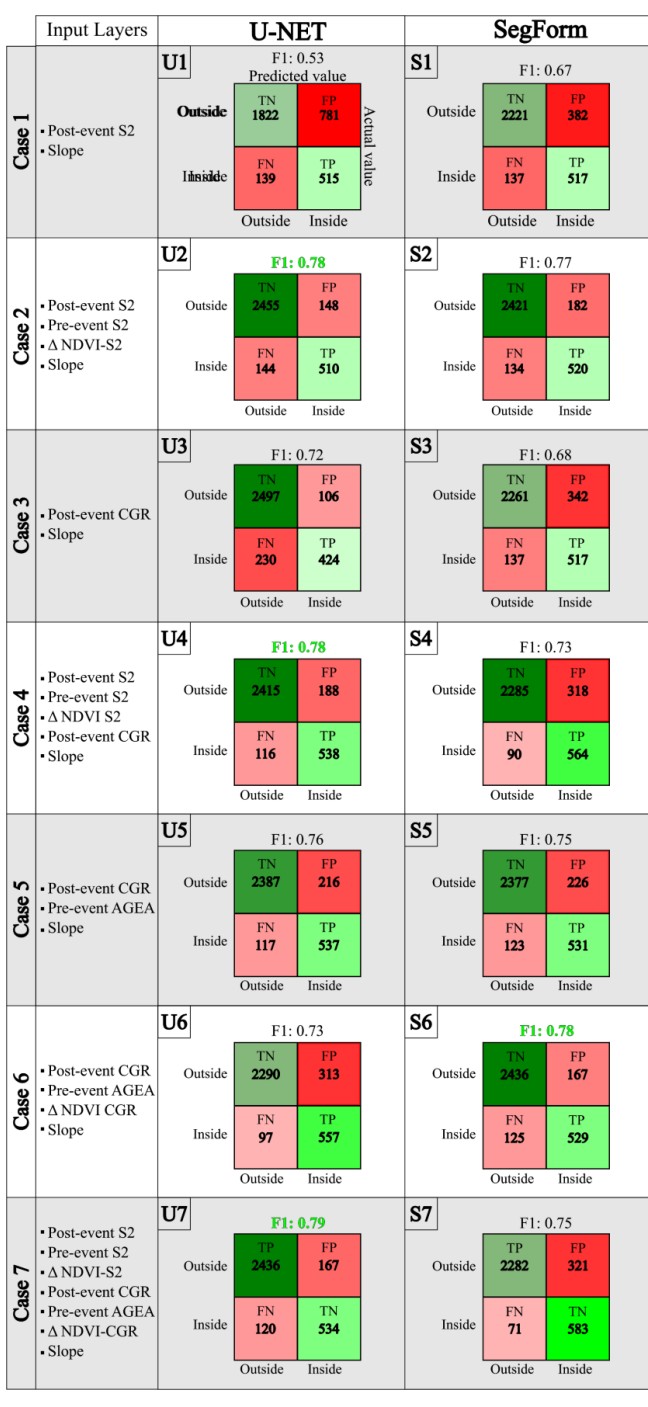

786

**Figure 11.** Comparison of buildings identified as at risk (within landslide boundaries or within a 20-meter buffer) by automated mapping methods and manual mapping (ground truth), showing the confusion matrices for all five cases evaluated.





## 6. Conclusion

In this work, we investigated the potential of automated landslide mapping to support rapid emergency response following the extreme meteorological events of May 2023 in Emilia-Romagna, Italy. Using a deep learning approach, we trained and tested two semantic segmentation models, U-Net and SegFormer, on high-resolution aerial, Sentinel-2 imagery, NDVI change maps, and slope data. The training was conducted solely in the municipality of Casola Valsenio, while model performance was assessed on three additional municipalities (Predappio, Modigliana, Brisighella), chosen for their geological settings and significant landslide occurrence.

The main findings of the study can be summarized as follows:

- **Training performance**: In the training area of Casola Valsenio, both models achieved high performance when provided with rich input combinations. U-Net reached F1 scores around 0.70, particularly when both high-resolution imagery and NDVI change maps were included (Figure 7). These results confirm that, when trained on spatially detailed and spectrally rich datasets, CNN-based models can closely replicate manually mapped landslide boundaries, including small and morphologically complex features.

- **Generalization to test areas**: When applied to unseen areas, both U-Net and SegFormer produced comparable F1 scores across Predappio, Modigliana, and Brisighella MA. While slight variations were observed, SegFormer occasionally performing better in certain configurations, there was no systematic advantage indicating stronger generalization. In Brisighella FAA, both models experienced a clear performance drop, highlighting the challenge of transferring models to geologically distinct terrains (Table 4, Figure 7). These results suggest that generalization is primarily constrained by lithological dissimilarity, rather than by architectural differences between CNN and transformer-based models.

- **Model comparison**: Despite their architectural differences, the two models achieved similar performance across all test areas. This convergence likely reflects the limited variability and discriminative power of the available input layers, suggesting that both architectures were able to extract the most relevant features from the data (Figure 7). Improving the variety and informativeness of the input layers could unlock further improvements, especially in geologically heterogeneous regions.

- **F1 vs. Expert Index**: The Expert Performance Index (EPI) was largely consistent with the quantitative F1 scores, reinforcing the validity of standard segmentation metrics for assessing model quality (Figure 10). Experts noted that the best-performing models also produced outputs that were more coherent from a geomorphological perspective, but they also identified recurring issues such as over segmentation, misclassification in agricultural areas, and false positives along riverbanks. These errors, while systematic, are easily corrected during manual review, making the model highly valuable as a starting point for rapid emergency mapping.

- **Lithology influence**: The inclusion of lithology as an additional input layer had limited effect when the model was applied to areas already represented in the training dataset. However, in Brisighella FAA, where lithologies were almost absent from the training set, both models performed poorly unless explicitly retrained. This highlights that effective integration of



lithological information into AI-based mapping requires more than adding it as an input
feature: the training dataset must include landslide polygons across a representative range of
lithological contexts. When the model was retrained on Brisighella FAA, the F1-score
increased, confirming that geological consistency in the training data significantly improves
generalization performance.
▪ **Exposure and damage detection**: The automated maps successfully identified a majority of
the buildings located within or near landslide polygons, reaching an F1 score of approximately
0.78 for building-at-risk detection (Figure 11). However, both false positives (e.g., buildings
wrongly marked as at risk) and false negatives (e.g., undetected at-risk buildings) were
frequent. This limits the reliability of automated exposure assessments, particularly in densely
populated or infrastructure-rich areas. More advanced exposure analysis methods that account
for slope direction, runout potential, and terrain morphology (e.g., BGC Engineering, 2010)
are needed to complement segmentation-based approaches.
Overall, automated landslide mapping offers substantial advantages in emergency scenarios. While
not a substitute for manual efforts, it provides a reliable baseline that can significantly accelerate the
production of actionable maps. With further refinement, such as full-resolution imagery (Wang et al.,
2022), improved tiling strategies (Abrahams et al., 2024), and data augmentation (Prakash et al.,
2021), these tools can become critical assets in disaster management workflows. Given the increasing
scale and frequency of climate-driven disasters, integrating AI-based approaches into civil protection
protocols could enhance both the speed and consistency of emergency responses. Deep learning
models, despite current limitations, represent a scalable solution to the growing demand for rapid,
accurate, and spatially detailed hazard information. By automatically addressing the most evident
features, these models reduce the burden on experts, allowing them to focus on verifying complex or
ambiguous regions. Nonetheless, manual validation remains essential, as AI outputs alone cannot yet
ensure sufficient reliability in sensitive contexts.

**Acknowledgements**
This study was carried out within the RETURN Extended Partnership and received funding from the
European Union Next-GenerationEU (National Recovery and Resilience Plan – NRRP, Mission 4,
Component 2, Investment 1.3 – D.D. 1243 2/8/2022, PE0000005)

**Disclosures and declarations**
Declaration of AI and AI-assisted technologies in the writing process
During the preparation of this work the authors used ChatGPT 4 (chat.openai.com) to enhance the
grammar and syntax, as well as to refine the sentence structure. All the content is original, and no
concepts, ideas, or interpretations were produced by this tool. After using this tool, the authors



reviewed and edited the content as needed and take full responsibility for the content of the
publication.

**Code and data availability**

The dataset used in this study is openly available at https://doi.org/10.5281/zenodo.13742643.
The code developed for the analyses is available from the authors upon reasonable request.

**Author contribution**

NDS performed the data analysis, developed the scripts, and wrote the manuscript.
GC contributed to the interpretation of the results and supervised the work.
DE contributed to the methodology and algorithmic framework.
ELP contributed to the methodology and algorithmic framework.
AC supervised the interpretation of the results.
MB conceived the study, supervised the analysis and manuscript preparation, and provided critical
review and guidance throughout the research process.

**Competing interests**

The authors declare they have no financial interests.

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
