# Peer review of "Rapid Landslide Mapping During the 2023 Emilia-Romagna Disaster: Assessing Automated"

_EGUsphere, 2025_

## Referee Comment (RC1)

**Review Report**

**Manuscript**: Rapid Landslide Mapping During the 2023 Emilia-Romagna Disaster: Assessing Automated Approaches with Limited Training Data

The study employed deep learning techniques based on convolutional neural network architectures (U-Net and SegFormer) to segment landslides across regions with distinct geological characteristics. The primary objective was to assess the effectiveness and limitations of automated landslide mapping in practical scenarios and to evaluate whether deep-learning approaches can reliably replace manual mapping. The results highlight several important aspects, including a comparison of models trained using different architectures and input layers. The discussion further examines the impact of incorporating a lithological layer into the model and presents an analysis of buildings potentially affected by landslide hazards. Despite the interesting approach to landslide segmentation, the manuscript presents structural and organizational weaknesses that at times hinder readability. A revision is therefore recommended to improve the clarity of the methodological framework and the presentation of results.

**Major Review**
**1. Introduction:**

The paragraph in lines 56–64, which reviews studies employing Convolutional Neural Networks (CNNs) and deep learning techniques for landslide mapping, should be strengthened by incorporating a broader range of relevant literature, particularly more recent studies. Expanding this section would provide a more comprehensive and up-to-date overview of current advances in deep learning–based landslide mapping and better contextualize the contribution of the present work.

The paragraph in lines 65–73 correctly notes that previous studies typically adopt large train-to-validation ratios (80:20 or 70:30), but it does not sufficiently clarify why this constitutes a limitation in real-time crisis scenarios. The authors should explicitly explain that, during an emergency, only a very limited portion of the affected area is usually mapped and annotated in the early hours or days, making large training sets unrealistic. Additional constraints, such as delays in acquiring cloud-free post-event imagery, the absence of recent pre-event data, and the operational need for rapid deployment, further restrict the amount and quality of training data available. The paragraph also states that training procedures often fail to represent the diversity of landslide types and geological conditions encountered in practice; however, this point would benefit from brief elaboration.

**3. Study Area**

The description of the study areas (Casola Valsenio, Brisighella, Modigliana, and Predappio) should be moved before the initial paragraphs of this section. These initial paragraphs discuss geological conditions and terrain variability that are only detailed later in the text, resulting in a forward reference that disrupts the narrative flow.

The section lists the number of landslides, their types, and relative percentages for each study area. A summary table compiling this information for all municipalities would greatly facilitate comparison and comprehension of the landslides among sites.

**4. Methods**

The paragraphs from lines 298–354 provide a comprehensive overview of the different data-availability scenarios (L1–L7) considered in the study; however, they are rather long and difficult to follow in their current form. The description alternates between data-availability assumptions, sensor characteristics, and computational constraints, which difficulties the understanding of the logical structure of the scenarios summarized in Table 2. Explicitly linking each paragraph to the corresponding cases in Table 2 would significantly improve readability.

Lines 301–303 state that, in Table 2, the availability of a slope map (L7) is assumed based on the free accessibility of global DEMs (e.g., SRTM, ASTER, ALOS, Copernicus DEM). However, earlier in lines 295–297, the slope map is described as being derived from the 1:5000 Regional Technical Map. This creates ambiguity regarding the actual source(s) of the L7 slope product. If global DEMs were also used, alone or in combination with the regional dataset, they should be explicitly described as part of the L7 product. Alternatively, if L7 is derived exclusively from the regional map, this should be clearly stated and the reference to global DEMs clarified accordingly.

**4.x Deep Learning Semantic Segmentation Models**

The section first introduces the models and only later describes the training areas and data preparation. It would be clearer to reorganize the content to follow a more logical workflow: data preparation, model training, and data evaluation (metrics). In addition, some paragraphs are difficult to follow and would benefit from improved clarity and structure. Including additional figures illustrating the model architectures would also help facilitate understanding.

**4.x Model Application**

The first two paragraphs discuss aspects related to the test area and would be more appropriately placed in the training–testing split section. Additionally, including a table showing the percentage of each test area within each lithological unit would improve clarity and facilitate understanding of how the models were evaluated.

**5 Results**

The second paragraph (563-573) should explicitly show the results and instead of using words like "highest" , "achieve results close to those of the top-performing models"..

In general the paragraphs do not explicitly state the values encountered by the models and some paragraphs should be rewrite to be more precise. I recommend reviewing this section completely.

**6 Discussion**

The section discusses some of the results; however, it does not adequately interpret them in terms of the spectral response of the mapped targets. In addition, certain results, such as the inclusion of a lithological layer and the assessment of buildings potentially at risk, are introduced without having been previously described in the Methods section, which makes the paragraph difficult to follow. To improve clarity and coherence, these methodological aspects should be clearly presented earlier, and portions of the discussion should be relocated to the Results section to enhance overall readability.

**Minor Review**

**Abstract**: Well written but slightly long. Can be more objective and tightened by removing methodological repetition.

**3. Study Area**

136-137: Readability would be improved by explicitly citing the name of the municipalities;
144: Text states that "about 25% of the recorded 80,997 landslides events", adding the quantity of landslides would also improve the comprehension of the impacted area;
Figure 2 - Increase the legend size;
193: Correct from "sandstone-rice" to "sandstone-rich";

**3. Methods**
Review the numbering of the sections
Figure 4: Sentinel images are with low contrast which difficult the visualization;
304-309 - Should also describe the product level that was used in the study.
320-354 - Text refers to cases (e.g case 1, case 2 ..) however this definition is not represented or cited in table 2.

**3.7 Evaluation Metrics and Expert Judgment**

Lines 498-501 where the F1 metric is described, the text says that the metric is useful for imbalanced problems, but this was not clearly discussed in the methodology, was the data used in the experiment imbalanced?

**5 Results**
First paragraph (556-562) could be adapted and turned into the legend of table 4, as it does not describe any result, this would improve readability.

Figure 7 - Legend states "case 1 - 7" but in the table it is referred as "U1", "S1", "U2", "S2", also inputting these names would facilitate the comprehension of the table. The size of the legends should be increased.

Figure 8 and 9 - Changing the colors used to highlight the TP and FP would improve the visualization.

Paragraphs from 628-655 highlight difficulties of the model in the segmentation over different areas, but figure 9 does not highlight those areas making it harder to follow the paragraph.

**6 Discussion**

Paragraph from 715-726 states that another model was trained to evaluate the impact of including lithology but no explanation was added to describe how this was done. The categorical layer was converted to dummy layers (one for each category) or was added as a categorical layer? Extra information would improve the comprehension

The discussion in lines 734–741 regarding the F1-score relies on comparisons with generic machine-learning benchmarks rather than on metrics reported in landslide-mapping studies. The analysis should instead be grounded in the context of landslide detection and segmentation literature. In particular, the manuscript should address why F1-scores in landslide mapping are often relatively low, discussing contributing factors such as class imbalance between landslide and non-landslide areas, the spatial heterogeneity of landslides, uncertainties in reference inventories, and the influence of image resolution and labeling quality.

**7 Conclusion**

This section should also address future research directions aimed at overcoming the limitations identified in the study.

---

## Author Comment (AC1)

Dear Editor, Associate Editor, and Reviewers,

Thank you very much for your useful comments and suggestions.

In this document, you will find a detailed explanation of the changes made to the original manuscript to meet your suggestions.

For the sake of clarity, we used the following text styles:

| | |
|---|---|
| *black, italics*: | reviewer comment |
| blue, plain text: | our reply |
| *red, italics:* | revised text |

Best regards

Nicola Dal Seno

Giuseppe Ciccarese

Davide Evangelista

Elena Loli Piccolomini

Alessandro Corsini

Matteo Berti

*Major Review*

*R1.1 - "1. Introduction:"*

*(a) The paragraph in lines 56–64, which reviews studies employing Convolutional Neural Networks (CNNs) and deep learning techniques for landslide mapping, should be strengthened by incorporating a broader range of relevant literature, particularly more recent studies. Expanding this section would provide a more comprehensive and up-to-date overview of current advances in deep learning–based landslide mapping and better contextualize the contribution of the present work.*

*(b) The paragraph in lines 65–73 correctly notes that previous studies typically adopt large train-to-validation ratios (80:20 or 70:30), but it does not sufficiently clarify why this constitutes a limitation in real-time crisis scenarios. The authors should explicitly explain that, during an emergency, only a very limited portion of the affected area is usually mapped and annotated in the early hours or days, making large training sets unrealistic. Additional constraints, such as delays in acquiring cloud-free post-event imagery, the absence of recent pre-event data, and the operational need for rapid deployment, further restrict the amount and quality of training data available. The paragraph also states that training procedures often fail to represent the diversity of landslide types and geological conditions encountered in practice; however, this point would benefit from brief elaboration.*

We thank the Reviewer for this valuable and constructive comment. In response, we substantially revised the Introduction to provide a more comprehensive, up-to-date, and operationally grounded overview of deep-learning based landslide mapping.

(a) First, we expanded the paragraph reviewing CNN and deep learning approaches by incorporating a broader range of recent and relevant literature (2020–2024). The revised text now includes studies addressing model generalisation, transferability across regions and events, systematic model comparisons, benchmark datasets for reproducibility, and recent hybrid CNN–Transformer architectures. These additions strengthen the state-of-the-art discussion and better contextualize the contribution of the present work within current advances in automated landslide mapping (e.g., Prakash et al., 2020; Ghorbanzadeh et al., 2022; Meena et al., 2023; Xu et al., 2024; Chen et al., 2023; Wu et al., 2024).

(b) Second, we revised the paragraph discussing training strategies to explicitly clarify why the large train-to-validation ratios commonly adopted in the literature (e.g., 80:20 or 70:30) represent a limitation in real-time emergency scenarios. We now state that, during an ongoing crisis, only a small fraction of the affected area is typically mapped and annotated in the first hours or days, making extensive training datasets unrealistic. We also added a brief discussion of additional operational constraints, including delays in acquiring cloud-free post-event imagery, limited or outdated pre-event reference data, and the need for rapid model deployment. Finally, we elaborated on the issue of representativeness by noting that early training samples often fail to capture the full diversity of landslide types and geological/geomorphological settings, which can negatively affect model generalisation during emergency response.

These revisions were implemented by jointly considering related comments raised by other reviewers, ensuring a coherent and consistent revision of the Introduction. The full revised Introduction is reported below for clarity.

*"Rapid and accurate landslide mapping over large areas is a challenging task, particularly in the context of regional-scale rainfall or earthquake events (Iverson et al. 2015; Casagli et al. 2016; Robinson et al., 2017; Holbling et al. 2017; Mondini et al., 2021). This task is crucial for implementing timely and effective response measures, assessing the extent of impact, and planning for recovery and mitigation efforts. The complexities of rapid landslide mapping vary depending on several factors including the extent of the affected area, the availability of cloud-free imagery, the precision needed for the mappings, and the difficulties of detecting landslides due to shadows, vegetation cover, or weak geomorphic evidences (Mondini et al. 2019; Amatya et al., 2023). Typically, these complexities are addressed by expert geologists through the visual interpretation of satellite or aerial imagery (Guzzetti et al., 2012; Scaioni et al., 2014; Ferrario and Livio 2023). These experts are skilled at identifying subtle variations in the terrain and can integrate complex contextual knowledge, including field reports, personal accounts, and specific geological information, to create highly accurate landslide maps. However, manual mapping is both time-consuming and inherently subjective, which are significant limitations during an emergency (Novellino et al. 2024).*

*Before the widespread adoption of deep learning, automated landslide mapping approaches evolved through several methodological stages. Early methods relied mainly on pixel-based techniques, such as thresholding of spectral indices or terrain parameters, which were simple to implement but highly sensitive to noise and illumination conditions. These approaches were later complemented by object-based image analysis (OBIA), which segments images into meaningful objects and classifies them based on spectral, spatial, and contextual features (Blaschke 2010; Martha et al. 2010). OBIA-based methods represented a substantial improvement by incorporating shape, texture, and neighborhood information, and were widely applied to landslide detection from high-resolution imagery. However, both pixel- and object-based approaches typically depend on manually designed rules or handcrafted features, limiting their transferability across different events, sensors, and geological settings.*

*More recently, machine-learning and deep-learning approaches have progressively replaced rule-based systems by enabling data-driven feature extraction and end-to-end learning. Automated landslide recognition techniques using convolutional neural networks (CNNs) have emerged as promising alternatives to manual mapping (Sameen and Pradhan 2019; Chen et al. 2018; Ye et al. 2019; Tang et al. 2022; Ji 2020). Numerous studies have demonstrated their effectiveness in real-world post-disaster scenarios. For example, Meena et al. (2021) applied a deep learning approach for rapid landslide mapping in India following extreme monsoon rainfall, while Prakash et al. (2021) introduced a generalized CNN framework designed to improve applicability across different geographic contexts. Prakash and Manconi (2021) further demonstrated the operational value of CNN-based mapping by rapidly delineating landslides triggered by severe storm events. Beyond individual case studies, recent research has focused on transferability, reproducibility, and systematic comparison of methods. Prakash et al. (2020) explicitly compared deep-learning models with traditional machine-learning approaches for landslide mapping from Earth observation data, highlighting the advantages of deep architectures in complex environments. In parallel, benchmark datasets such as Landslide4Sense (Ghorbanzadeh et al. 2022), HR-GLDD (Meena et al. 2023), and the CAS Landslide Dataset (Xu et al. 2024) have been released to promote standardized evaluation across multiple events, sensors, and geographic regions. These efforts have significantly advanced the field by enabling more robust inter-comparisons and by revealing persistent challenges related to class imbalance, spatial heterogeneity, and generalization across geological domains.*

*In parallel with CNN developments, transformer-based architectures have recently been introduced in landslide mapping to better capture long-range spatial dependencies and multi-scale contextual information. Unlike CNNs, which primarily exploit local convolutional kernels, transformers rely on self-attention mechanisms that allow each pixel (or patch) to attend to a broader spatial context. This property is particularly relevant for landslide detection, where the geomorphological setting, slope connectivity, and spatial coherence of failures extend beyond local neighborhoods. Hybrid CNN–Transformer models and pure transformer-based architectures have shown promising results in improving robustness and generalization across heterogeneous landscapes (Chen et al. 2023; Wu et al. 2024). These characteristics are especially attractive for rapid response applications, where models must be deployed with limited training data and applied to large, spatially diverse areas under operational constraints.*

*Despite these advances, a substantial gap remains between methodological developments and their practical deployment during emergency response. Most deep-learning studies rely on relatively large and well-balanced training datasets, often adopting train-to-validation ratios such as 80:20 or 70:30 (Meena et al. 2023). In contrast, during an ongoing crisis only a small fraction of the affected area is typically mapped and annotated in the first hours to days, making extensive training datasets unrealistic. This limitation is often exacerbated by delays in acquiring cloud-free post-event imagery, inconsistencies in pre-event reference data, and the operational need to deploy models rapidly with minimal manual intervention. Furthermore, early training samples are frequently not representative of the full variability of landslide processes and environmental conditions, including different landslide types, lithologies, land-cover patterns, and illumination or shadow effects. As a result, models trained on limited and potentially biased early annotations may generalize poorly when applied to the broader affected region.*

*In this study, we utilize the landslide inventory from the May 2023 crisis in the Emilia-Romagna region of Italy (Berti et al. 2025) to assess the effectiveness and limitations of automated mapping algorithms in a practical setting. We specifically focus on the performance of the U-Net and SegFormer algorithms in identifying landslides triggered by the event, despite the challenges of very limited training data. The study also considers the variety of landslide types and geological conditions prevalent in the area, which impacted the disaster response. The main goal of this research is to determine whether advanced deep-learning methods can effectively replace manual mapping in managing large-scale landslide disasters, thus bridging the gap between academic research and real-world application in emergency management."*

*"**References***

*Blaschke, T. (2010). Object based image analysis for remote sensing. ISPRS Journal of Photogrammetry and Remote Sensing, 65(1), 2–1. https://doi.org/10.1016/j.isprsjprs.2009.06.004*

*Chen, X., Liu, M., Li, D., Jia, J., Yang, A., Zheng, W., & Yin, L. (2023). Conv-trans dual network for landslide detection of multi-channel optical remote sensing images. Frontiers in Earth Science, 11, 1182145. https://doi.org/10.3389/feart.2023.1182145*

*Ghorbanzadeh, O., Xu, Y., Ghamisi, P., Kopp, M., & Kreil, D. (2022). Landslide4Sense: Reference Benchmark Data and Deep Learning Models for Landslide Detection. IEEE Transactions on Geoscience and Remote Sensing, 60. https://doi.org/10.48550/arXiv.2206.00515*

*Martha, T. R., Kerle, N., Jetten, V., van Westen, C. J., & Kumar, K. V. (2010). Characterising spectral, spatial and morphometric properties of landslides for semi-automatic detection using object-oriented methods. Geomorphology, 116, 24–36. https://doi.org/10.1016/j.geomorph.2009.10.004*

*Prakash, N., Manconi, A., & Loew, S. (2020). Mapping Landslides on EO Data: Performance of Deep Learning Models vs. Traditional Machine Learning Models. Remote Sensing, 12(3), 346. https://doi.org/10.3390/rs12030346*

*Wu, L., Liu, R., Ju, N., Zhang, A., Gou, J., He, G., & Lei, Y. (2024). Landslide mapping based on a hybrid CNN-transformer network and deep transfer learning using remote sensing images with topographic and spectral features. International Journal of Applied Earth Observation and Geoinformation, 126, 103612. https://doi.org/10.1016/j.jag.2023.103612*

*Xu, Y., Ouyang, C., Xu, Q., et al. (2024). CAS Landslide Dataset: A Large-Scale and Multisensor Dataset for Deep Learning-Based Landslide Detection. Scientific Data, 11, 12. https://doi.org/10.1038/s41597-023-02847-z"*

***R1.3 - "3. Study Area"***

*(a) The description of the study areas (Casola Valsenio, Brisighella, Modigliana, and Predappio) should be moved before the initial paragraphs of this section. These initial paragraphs discuss geological conditions and terrain variability that are only detailed later in the text, resulting in a forward reference that disrupts the narrative flow.*

We thank the Reviewer for this helpful comment. In response, we reorganized the "Study area" section by moving the detailed description of the four municipalities (Casola Valsenio, Brisighella, Modigliana, and Predappio) to the beginning of the section. This reordering removes the forward reference to geological and terrain characteristics that were previously introduced before the study areas were described.

*(No changes in content were required; the modification consists solely of a reorganization of the section structure.)*

*(b) The section lists the number of landslides, their types, and relative percentages for each study area. A summary table compiling this information for all municipalities would greatly facilitate comparison and comprehension of the landslides among sites.*

We agree with the Reviewer. We added a new summary table "Table 1" compiling the total number of landslides and the breakdown by landslide type for all four municipalities, in order to facilitate comparison across sites. The table is derived from the official RER2023 inventory (Berti et al., 2025), which is openly available as a Zenodo vector dataset (Pizziolo et al., 2024).

*"Table 1. Summary of the landslides mapped in the four study municipalities after the May 2023 Emilia-Romagna event, including total counts and breakdown by landslide type. DS = debris slides; DF = debris flows; DF1 = long-runout debris flows; DF2 = limited-runout debris flows; ES = earth slides; EF = earth flows; RS = rock-block slides. Data are from the RER2023 landslide inventory (Berti et al., 2025) and its open-access release (Pizziolo et al., 2024)."*

| Municipality | Tot. Landslides | Debris Slides (DS) | Debris Flows (DF) | Earth Slides (ES) | Earth Flows (EF) | Rock Slides (RS) |
|---|---|---|---|---|---|---|
| Casola Valsenio | 5572 | 4543 | 789 | 6 | 11 | 110 |
| Brisighella | 6342 | 3890 | 999 | 835 | 593 | 25 |
| Modigliana | 6974 | 5357 | 1306 | 152 | 90 | 69 |
| Predappio | 6832 | 5476 | 750 | 426 | 117 | 31 |

***R1.4 – "4. Methods"***

*(a) The paragraphs from lines 298–354 provide a comprehensive overview of the different data-availability scenarios (L1–L7) considered in the study; however, they are rather long and difficult to follow in their current form. The description alternates between data-availability assumptions, sensor characteristics, and computational constraints, which difficulties the understanding of the logical structure of the scenarios summarized in Table 2. Explicitly linking each paragraph to the corresponding cases in Table 2 would significantly improve readability.*

*(b) Lines 301–303 state that, in Table 2, the availability of a slope map (L7) is assumed based on the free accessibility of global DEMs (e.g., SRTM, ASTER, ALOS, Copernicus DEM). However, earlier in lines 295–297, the slope map is described as being derived from the 1:5000 Regional Technical Map. This creates ambiguity regarding the actual source(s) of the L7 slope product. If global DEMs were also used, alone or in combination with the regional dataset, they should be explicitly described as part of the L7 product. Alternatively, if L7 is derived exclusively from the regional map, this should be clearly stated and the reference to global DEMs clarified accordingly.*

(a) We thank the reviewer for this helpful comment. We revised Section 3.1 to improve readability by explicitly linking each data-availability scenario to the corresponding cases in Table 3 and by reorganising the text following a clear progression from data-poor to data-rich conditions.

(b) We also clarified the source of the slope layer (L7), which was ambiguous in the previous version. We now state explicitly that, in this study, L7 was derived from the Emilia-Romagna regional 5 × 5 m DTM (based on the 1:5000 Regional Technical Map). The mention of global DEMs (e.g., SRTM, ASTER, ALOS, Copernicus DEM) is retained only to justify the general assumption that a slope product is typically obtainable in emergency contexts, even when regional datasets are not available.

*"These layers were combined to define seven data-availability scenarios (cases 1-7 in Table 3), each reflecting realistic operational conditions following a landslide-triggering event.*

*In all scenarios, the availability of a slope map (L7) was assumed. In the present study, L7 was derived exclusively from the regional 5 × 5 m DTM. However, this assumption reflects a more general operational condition, as slope information can be readily derived from freely available global digital elevation models (e.g., SRTM, ASTER, ALOS, Copernicus DEM), which are commonly accessible even in data-scarce emergency contexts.*

*Case 1 (Table 3) represents the most data-limited scenario, where only post-event Sentinel-2 imagery (L1) is available. Owing to its global coverage, short revisit time, and free accessibility, Sentinel-2 data often constitute the primary information source immediately after large-scale disasters (Wasowski et al., 2014; Yang et al., 2019; Ban et al., 2020).*

*Case 2 considers situations in which pre-event Sentinel-2 imagery (L2) is also accessible, allowing the computation of an NDVI change map (L3). While this condition is frequently met, prolonged cloud cover or strong seasonal vegetation variability may prevent the availability of suitable pre-event images.*

*Cases 3 and 4 describe scenarios in which high-resolution post-event imagery is acquired, either alone (case 3) or in combination with Sentinel-2 data (case 4). Such data may originate from very-high-resolution satellite*

*systems (e.g., WorldView, Pleiades, SkySat) or dedicated aerial surveys and substantially improve the detection of small or geomorphically subtle landslides.*

*Cases 5 and 6 address less common situations in which high-resolution imagery is available both before and after the event, optionally complemented by a high-resolution NDVI change map. The effectiveness of these scenarios depends strongly on the temporal proximity of the pre-event imagery to the triggering event.*

*Finally, case 7 represents the most data-rich configuration, combining all available layers (L1–L7).*

*Although the post-event (L4), pre-event (L5), and derived NDVI change (L6) aerial datasets were originally available at 0.2 m spatial resolution, their use at native resolution resulted in systematic out-of-memory errors during training and inference due to the large size of the input tensors. To ensure computational stability, were therefore downsampled to 2 m, representing a practical compromise between spatial detail and computational feasibility for regional-scale mapping. By contrast, Sentinel-2 layers (L1–L3) and the slope layer (L7) were resampled to the same 2 m grid solely to ensure pixel-wise alignment when used together with the aerial datasets; this step does not add information beyond their native resolution and was performed for operational consistency."*

**R1.5 – "4.x Deep Learning Semantic Segmentation Models"**

*The section first introduces the models and only later describes the training areas and data preparation. It would be clearer to reorganize the content to follow a more logical workflow: data preparation, model training, and data evaluation (metrics). In addition, some paragraphs are difficult to follow and would benefit from improved clarity and structure. Including additional figures illustrating the model architectures would also help facilitate understanding.*

We thank the reviewer for this valuable suggestion. Section 4 has been thoroughly reorganised to follow a clearer and more logical workflow consistent with standard deep-learning pipelines. The revised structure now introduces data preparation and the training–testing strategy before describing the model architectures, followed by model application and evaluation metrics.

In addition, the descriptions of the U-Net and SegFormer models were revised to improve clarity and internal structure by separating conceptual aspects from implementation details and reducing redundancy. To further facilitate understanding, a new figure (Fig. 6) illustrating the U-Net and SegFormer architectures has been added, as suggested, and is explicitly referenced in the text.

*"4.4 Data preparation and training-testing design*

*To reproduce operational conditions typical of post-event emergency mapping, the deep-learning workflow was designed to prioritise spatial generalisation rather than random pixel-level splitting. All input layers described in Section 3.1 were harmonised to a common spatial resolution and grid prior to training.*

*The training area (Casola Valsenio municipality) was subdivided into regular 1 × 1 km tiles (Fig. 5) to define consistent spatial sampling units and to reduce spatial autocorrelation effects. When resampled to a spatial*

resolution of 2 m, each tile corresponds to 512 × 512 pixels in the CGR imagery. Individual tiles typically include a large number of mapped landslides (often exceeding 50), providing a robust representation of landslide morphology and surrounding land cover.

Instead of conventional random splits (e.g. 70/30 or 80/20; Hastie et al., 2009), we adopted a strategy tailored to emergency response scenarios. The models were trained exclusively on data from Casola Valsenio and subsequently applied, without retraining, to three neighbouring municipalities, Predappio, Modigliana, and Brisighella, for independent testing. This setup simulates a realistic operational context in which a model trained on a limited reference area is rapidly deployed to map other affected regions.

Within Casola Valsenio, a stratified random sampling strategy was used to partition the tiles into training, validation, and internal testing subsets. Out of 97 tiles, 60 (62%) were assigned to training, 15 (15%) to validation, and 22 (23%) to internal testing (Fig. 5). External evaluation was then conducted on all tiles from the three remaining municipalities, providing a spatial hold-out test to assess cross-municipality generalisation. This tile-based spatial splitting yields a strongly imbalanced pixel-level segmentation problem, because landslide pixels represent only a small fraction of the mapped area. In Casola Valsenio (training area), landslide pixels account for ~ 4% of the total pixels (non-landslide:landslide ratio ~24:1). A comparable imbalance characterises the external test areas, with ~6% landslide pixels in Predappio (~ 16:1), ~7.6% in Modigliana (~12:1), and ~5% in Brisighella FMA (~19:1). The most extreme imbalance occurs in Brisighella FAA, where landslide pixels represent ~2% of the pixels (~48:1). The implications of this class imbalance for model evaluation are discussed in Section 4.7.

**4.5 Deep Learning Semantic Segmentation Models**

To evaluate the performance of different deep-learning paradigms under variable data-availability scenarios (Table 3), two semantic segmentation architectures were implemented: a convolutional encoder–decoder network (U-Net) and a Transformer-based segmentation model (SegFormer). Schematic representations of both architectures are provided in Fig. 6 to facilitate understanding and comparison.

**4.5.1 U-Net**

U-Net is a convolutional neural network originally developed for biomedical image segmentation (Ronneberger et al., 2015) and subsequently adopted in a wide range of geomorphological and Earth-observation applications. Its encoder–decoder structure enables the extraction of contextual information while preserving fine-scale spatial detail, a key requirement for accurate landslide delineation. The suitability of U-Net for landslide mapping has been demonstrated in several recent studies (Meena et al., 2021, 2022; Ghorbanzadeh et al., 2022, 2023; Nava et al., 2022).

*The architecture consists of a contracting encoder path and an expanding decoder path. In the encoder, spatial resolution is progressively reduced through convolutional and max-pooling layers to capture high-level contextual features. In the decoder, feature maps are upsampled using transposed convolutions and concatenated with corresponding encoder features via skip connections, which preserve spatial information and improve boundary localisation (Fig. 6a).*

*In this study, U-Net was implemented manually in Python and configured for binary semantic segmentation (landslide vs non-landslide). The network employed two downsampling scales, each comprising two convolutional layers. The initial number of filters was set to 64 and increased with network depth to capture progressively more complex features.*

*Training was performed using the Adam optimiser (Kingma and Ba, 2017) with a learning rate of $1 \times 10^{-4}$ and Dice loss, for up to 300 epochs. Early stopping with a patience of 20 epochs was applied based on validation loss. All experiments were conducted using Python 3.8.18, TensorFlow 2.5.0, and CUDA 11.2.*

*4.5.2 SegFormer*

*SegFormer is a Transformer-based architecture specifically designed for efficient semantic segmentation of high-resolution and multispectral imagery (Xie et al., 2021). It addresses several limitations of earlier Vision Transformer models, including high computational cost, dependence on positional encodings, and inefficient multi-scale feature aggregation (Dosovitskiy et al., 2020).*

*The SegFormer architecture comprises two main components (Xie et al., 2021): (i) a hierarchical Transformer encoder that captures features at multiple spatial scales using efficient self-attention without positional encodings and Mix-FFN blocks with depth-wise convolutions, and (ii) a lightweight all-MLP decoder that fuses multi-level encoder features for dense per-pixel prediction (Fig. 6b). The absence of positional encodings allows the model to handle variable input resolutions without interpolation, which is advantageous for heterogeneous landslide imagery.*

*We adopted the MiT-B0 variant (Xie et al., 2021), with hidden sizes [32, 64, 160, 256], encoder depths [2, 2, 2, 2], and a decoder hidden size of 256. Although SegFormer was originally designed for three-channel RGB imagery, we exploited the flexibility of the SegformerForSemanticSegmentation implementation provided by the Hugging Face Transformers library (Wolf et al., 2020) to accommodate multi-channel inputs.*

*Training was performed using Cross-Entropy loss and the Adam optimiser with a learning rate of $1 \times 10^{-3}$, for up to 300 epochs, with early stopping based on validation loss (patience = 20 epochs). Computations were carried out using Python 3.8.10, PyTorch 1.9.0 (Paszke et al., 2019), and CUDA 11.1.*

[Figure]

***Figure 6.*** *Conceptual overview of the deep-learning semantic segmentation architectures used in this study. (a) U-Net encoder–decoder architecture with skip connections, illustrating the contracting path for contextual feature extraction and the expanding path for precise spatial localisation (adapted from Ronneberger et al., 2015). (b) SegFormer architecture, composed of a hierarchical Transformer encoder (MiT) with overlapping patch embeddings and efficient self-attention, and a lightweight all-MLP decoder for multi-scale feature fusion and dense per-pixel prediction (adapted from Xie et al., 2021)."*

***R1.6 – "4.x Model Application"***

*The first two paragraphs discuss aspects related to the test area and would be more appropriately placed in the training–testing split section. Additionally, including a table showing the percentage of each test area within each lithological unit would improve clarity and facilitate understanding of how the models were evaluated.*

We agree with the reviewer and have revised the manuscript accordingly. The paragraphs describing the spatial subdivision of the study area and the training–testing design have been moved to the training–testing split section, where they are now introduced prior to the model descriptions. This change improves the logical flow of the methodology and avoids redundancy within the Model Application section.

Furthermore, a new table (Table 4) has been added to summarise the number of 1 km² tiles used in each municipality, their role in the training, validation, and testing phases, and their distribution across the main

lithological units (Marnoso-Arenacea Formation and Pliocene Blue Clays). This table provides a quantitative overview of the geological contexts represented in the test areas and supports a clearer interpretation of the model evaluation and transferability.

*"4.6 Model application and geological transferability*

*Following training, the models were applied to the municipalities of Predappio, Modigliana, and Brisighella. Although ground-truth landslide inventories were available for these areas (Fig. 1a), they were not used during training and were reserved exclusively for model evaluation.*

*A key limitation of the training dataset is the predominance of the Marnoso-Arenacea Formation (FMA) in Casola Valsenio (Fig. 3a). To explicitly assess geological transferability, the Brisighella municipality was subdivided into two distinct lithological domains: one dominated by the Marnoso-Arenacea Formation (Brisighella FMA) and one characterised by Pliocene Blue Clays (Brisighella FAA; Fig. 3d).*

*The FAA domain is dominated by earth flows (EF) and earth slides (ES), typically developed within fine-grained, clay-rich materials and often associated with badland-like morphologies. These landslides generally exhibit elongated shapes, lobate toes, and sparse or degraded vegetation cover, reflecting repeated reactivation processes (Fig. 7a–b). In contrast, landslides in the FMA domain, both in Brisighella and in the training area, are mainly debris slides (DS) and debris flows (DF) triggered within thin colluvial layers overlying flysch bedrock. These failures tend to be rapid, with well-defined scarps and runout zones, and commonly affect densely vegetated slopes (Fig. 7c–d).*

*By explicitly separating these lithological contexts, we evaluated whether models trained solely on FMA-type landslides could generalise to areas characterised by fundamentally different landslide processes, geometries, and surface expressions. A summary of the spatial extent of each lithological unit within the test municipalities is provided in Table 4 to support interpretation of the evaluation results.*

*Table 4. Overview of the 1 km² tiles used for model training, validation, and testing across the analysed municipalities. The table reports the number of tiles assigned to each dataset split and their distribution within the main lithological units, namely the Marnoso-Arenacea Formation (FMA) and the Pliocene Blue Clays (FAA). External testing was performed on 100% of the tiles from Predappio, Modigliana, and Brisighella to assess cross-municipality and cross-lithology model generalisation.*

| Municipality | Total tiles [1 km²] | Training | Validation | Internal test | External test | FMA 'Unit 7' | FAA 'Unit 1' |
|---|---|---|---|---|---|---|---|
| Casola Valsenio | 97 | 60 | 15 | 22 | – | 97 | 0 |
| Predappio | 90 | – | – | – | 90 | 75 | 15 |
| Modigliana | 105 | – | – | – | 105 | 99 | 6 |
| Brisighella FMA | 116 | – | – | – | 116 | 116 | 0 |
| Brisighella FAA | 80 | – | – | – | 80 | 0 | 80 |

***R1.7 – "5 Results"***

*The second paragraph (563-573) should explicitly show the results and instead of using words like "highest", "achieve results close to those of the top-performing models".*

*In general the paragraphs do not explicitly state the values encountered by the models and some paragraphs should be rewrite to be more precise. I recommend reviewing this section completely.*

We thank the reviewer for this insightful comment and fully agree with the observation. Following the reviewer's suggestion, Section 4.1 (*Model results and performance*) and Section 4.2 (*Expert judgement*) were completely revised to ensure that all results are reported in a clear, objective, and quantitative manner.

In the revised manuscript, qualitative expressions have been replaced by explicit numerical values, ranges, and differences in F1-score, IoU, and Expert Performance Index (EPI), consistently referenced to Tables and Figures. The revised text now strictly adheres to a results-oriented structure, avoiding interpretative statements and ensuring full traceability of each claim to the corresponding quantitative evidence (Table 5; Figs. 8–11).

**"5. Results**

*5.1 Model results and performance*

*The results of the analysis are reported in Table 5, which summarizes the F1 and IoU scores obtained by comparing the automated landslide maps generated by the two deep-learning models (U = U-Net; S = SegFormer) with the manually mapped reference inventory. Seven combinations of input layers (cases 1-7; Table 1) were tested, representing progressively richer input information. The models were trained using data from Casola Valsenio and subsequently applied to four independent test areas: Predappio, Modigliana, Brisighella FMA, and Brisighella FAA. In total, 56 automated landslide maps were produced (7 input configurations × 4 test areas × 2 models).*

*Across all municipalities, the highest performance was obtained by the most information-rich configurations, particularly U7 and U6 for U-Net and S7 for SegFormer. In Casola Valsenio, the best configuration reached F1 = 0.73 and IoU = 0.57 (U7), while in Predappio, Modigliana, and Brisighella FMA the highest F1-scores were consistently around 0.60–0.63 (e.g., U7 = 0.56–0.63; S7 = 0.60–0.61). Performance was systematically lower in Brisighella FAA, where the best configurations reached F1 = 0.53 and IoU = 0.36 (U6/U7), and F1 = 0.52 and IoU = 0.35 (S7) (Table 5).*

*Despite relying on a reduced set of inputs, the Sentinel-2-based configuration S2 (post-event Sentinel-2 plus Sentinel-2 NDVI change) achieved competitive results across municipalities. For instance, in Casola Valsenio S2 reached F1 = 0.62 and IoU = 0.45, compared with F1 = 0.73 and IoU = 0.57 for the best-performing configuration (U7). In Predappio, Modigliana, and Brisighella FMA, S2 yielded F1-scores between 0.54 and 0.57 (IoU = 0.37–0.40), which are close to the values obtained by the strongest configurations (typically within ~0.03–0.06 in F1 and ~0.03–0.06 in IoU). In Brisighella FAA, S2 remained among the most stable configurations (F1 = 0.47; IoU = 0.31), whereas some intermediate cases showed marked drops (e.g., U3: F1 = 0.29; IoU = 0.17).*

*Overall, increasing the number of input layers resulted in measurable but limited performance improvements. The largest gain was associated with the inclusion of NDVI change information (case 2 vs. case 1), while*

*subsequent additions of high-resolution inputs generally produced smaller incremental changes. These results indicate that streamlined configurations based on Sentinel-2 data (case 2), optionally complemented by high-resolution imagery, can achieve segmentation performance close to that obtained using the full set of available inputs (Table 5).*

*Figure 8 illustrates the evolution of F1-scores across the seven input configurations for each municipality. For both models, the inclusion of NDVI change information (case 2) led to an increase in F1-score relative to case 1. In Casola Valsenio, F1 increased from 0.56 to 0.70 for U-Net and from 0.47 to 0.54 for SegFormer. In Modigliana and Brisighella FAA, the corresponding increases were approximately +0.15 to +0.16 for U-Net. Beyond case 2, U-Net showed gradual increases in F1-score of 0.01-0.04 per configuration up to case 7, whereas SegFormer exhibited smaller variations, generally within ±0.02, indicating a lower sensitivity to further input enrichment.*

*Model performance varied across municipalities. In Casola Valsenio, U-Net F1-scores increased from 0.56 (U1) to 0.73 (U7), while SegFormer ranged from 0.47 (S1) to 0.66 (S7). In Predappio, Modigliana, and Brisighella FMA, U-Net F1-scores were between 0.55 and 0.63, whereas SegFormer values remained more stable, typically between 0.55 and 0.60. Brisighella FAA consistently showed the lowest performance for both models: U-Net F1-scores ranged from 0.29 (U3) to 0.53 (U7), and SegFormer values ranged from 0.32 to 0.52, with no systematic improvement beyond case 2. IoU values exhibited the same spatial pattern, decreasing from 0.35-0.46 in FMA-dominated areas to 0.17-0.36 in the FAA domain.*

*Figures 9 and 10 complement the quantitative results by providing a visual interpretation of model performance under contrasting conditions. Figure 9 illustrates representative cases in which both models performed well, showing a strong spatial agreement between the manually mapped inventory and the automated outputs. In these panels, the overlap between the reference landslide map (blue) and the predicted map (yellow) produces a beige colour, corresponding to true positives and indicating accurate landslide delineation. These examples mainly refer to debris slides and debris flows occurring in well-illuminated areas with widespread vegetation removal, where spectral and textural changes are pronounced.*

*Figure 10, by contrast, focuses on more challenging scenarios, where the models exhibited recurrent error patterns. In Brisighella FMA, false positives (yellow polygons) are frequently mapped along riverbanks and on recently constructed buildings, likely associated with flood-related sediment redistribution and strong spectral contrasts. The most critical conditions are observed in Brisighella FAA, where widespread false positives occur within badland morphologies developed on Blue Clay formations, highlighting the limited transferability of models trained predominantly on flysch-dominated terrains. In Modigliana, false positives (yellow polygons) are often observed in cultivated fields, where agricultural disturbance produces spectral signatures similar to recent landslides. Finally, in Predappio, false negatives (blue polygons) dominate in shadowed slopes, where reduced illumination and altered colour profiles limit the detectability of landslide features.*

*By explicitly contrasting successful detections (Figure 9) with failure modes (Figure 10), these visual examples clarify how different surface conditions, illumination effects, and lithological settings influence model performance, supporting the quantitative trends reported in Table 5 and Figure 8.*

*5.2 Expert judgement*

*Figure 11 compares the quantitative performance (F1-score) of all model configurations with the Expert Performance Index (EPI) derived from the independent assessment of three experts (Section 3.7). The EPI values reflect the severity scores assigned to seven recurrent mapping errors (E1–E7), which were aggregated and normalized to obtain a single index per model configuration.*

*Across configurations, EPI values show the same overall ranking observed for the F1-scores (Fig. 11). While the F1-scores span a relatively narrow interval across cases (Table 5), the EPI values exhibit a comparable monotonic increase from simpler to more information-rich input configurations, indicating that the expert-based assessment captures the same performance gradient. In particular, configuration U3 is associated with one of the lowest EPI values and one of the lowest F1-scores, whereas U7 consistently falls within the top-performing group in both metrics (Fig. 11).*

*Differences among individual experts are evident for some intermediate configurations. Specifically, the highest EPI score for Expert 1 is assigned to S5, for Expert 2 to U2, and for Expert 3 to U4, despite these configurations not being uniformly ranked at the top by the other experts or by the F1-scores. However, when expert scores are averaged (mean EPI; dashed line in Fig. 11), the resulting trend closely matches the F1-score curve, reducing the influence of individual preferences and providing a more robust overall assessment of map quality."*

**R1.8 – "6 Discussion"**

*(a) The section discusses some of the results; however, it does not adequately interpret them in terms of the spectral response of the mapped targets.*

We have addressed this comment by adding an explicit interpretation of the results in terms of the spectral response of the mapped targets. In particular, we introduced a dedicated subsection (Section 6.2) supported by Figure 14, where we compare the spectral signatures of representative classes (riverbanks affected by flooding, landslides, ploughed fields, and shadowed areas). We then link the observed spectral similarities/differences to the recurrent error patterns in the mapping outputs (Figure 10) and to the models' channel importance (Figure 13), providing a more physically grounded explanation of false positives and false negatives.

*"6. Discussion*

*6.1 Models comparison*

*One of the key observations from this study concerns the differing sensitivity of U-Net and SegFormer to input data combinations, particularly in the context of emergency response mapping. U-Net's performance varies significantly depending on the layers used, while SegFormer yields more consistent results across different configurations. This contrast is likely rooted in their architectural differences. U-Net, designed for pixel-wise segmentation, relies heavily on the spatial and spectral characteristics of the input data, making it more sensitive to the quality and availability of supplementary layers such as pre-event imagery or NDVI change maps. SegFormer, being a transformer-based model, is better equipped to capture global context, thus reducing dependency on specific input combinations and improving robustness under complex geological conditions.*

*This architectural contrast becomes even more evident when models are transferred across different geological domains. As shown in the results (Figure 8), U-Net exhibits a marked decline in performance when moving from areas geologically similar to the training site (e.g., Predappio, Modigliana) to Brisighella FAA, where lithological conditions differ significantly. This suggests that U-Net has limited generalization capacity beyond the Marnoso-Arenacea Formation (FMA), and struggles in fine-grained, clay-rich terrains like those found in the FAA domain. SegFormer, although generally more stable, also underperforms in Brisighella FAA, highlighting the broader challenge of transferring models across heterogeneous geological settings.*

*The feature importance values for U-Net and SegFormer (Figure 13) reveal notable differences in how each model prioritizes the input channels. Upon examining the results, we can observe the following key points:*

- *SegFormer places the highest importance on the post-event CGR channels, particularly the Red and NIR bands, followed by Sentinel-2 Red and Sentinel-2 NIR. This makes sense in the context of post-event analysis, as Red and NIR bands are sensitive to vegetation and surface changes, which are crucial for detecting variations in the landscape after a landslide event. The reliance on these bands is likely due to their ability to capture the shift in vegetation cover (from green to beige or brown) after a disturbance, which can be a key indicator of landslide activity. The CGR (post) channels are particularly effective at highlighting these changes, making them essential for SegFormer's decision-making process.*

- *U-Net, on the other hand, follows a similar pattern but with lower variability in feature importance across different channels. While the CGR (post) channels remain important, U-Net seems to show a more uniform distribution of feature importance across the input channels, indicating that the model is less sensitive to specific features like Red and NIR. This could be a result of the model's pixel-wise segmentation approach, which might focus more on spatial relationships within the image rather than relying heavily on spectral differences between the bands.*

[Figure]

***Figure 13.*** *Feature importance values for U-Net and SegFormer across different input channels. The bar chart compares the relative importance assigned to each channel by both models. The blue bars represent U-Net, while the orange bars represent SegFormer. The analysis shows that SegFormer tends to assign higher importance to the post-event CGR channels, particularly the Red and NIR bands, while U-Net exhibits more uniform importance across the channels, with a slightly more localized focus. This difference in feature importance highlights the varying strengths of the models in utilizing spectral information.*

*Both models exhibit a similar trend in emphasizing the post-event data, particularly the CGR (post) channels. However, SegFormer seems to make use of a wider range of channels, including Sentinel-2 Red and Sentinel-2 NIR, which suggests that SegFormer is more adept at capturing a broader, global context. This capability allows it to effectively integrate multiple data sources. In contrast, U-Net appears to rely more heavily on the specific channels it uses, with its performance being more sensitive to the quality and selection of those channels.*

*This difference in how the models handle feature importance reflects their underlying architectural differences. SegFormer, with its transformer-based design, is able to capture larger-scale patterns and dependencies within the data, giving it the flexibility to work with diverse and comprehensive input combinations. On the other hand, U-Net's pixel-wise segmentation approach tends to focus on more localized patterns, which can make it less adaptable when handling a variety of input channels.*

*Nevertheless, both models encountered certain challenges that influenced their performance, which we will now address in the following section.*

*6.2 Challenges Scenarios*

*The models in this study encountered several challenging scenarios related to the environmental conditions of the test sites. These challenges, including riverbank areas affected by flooding, geological variations, plowed land, and shadowed regions, directly influenced the models' ability to detect landslides accurately. Spectral analysis of each scenario (Figure 14) reveals distinct patterns that explain the models' performance limitations, while Figure 10 displays the corresponding mapping results with recurrent error patterns.*

*a) Riverbank Areas Post-Flooding*

*The May 2023 flood events caused spectral signatures in riverbank areas that partially overlap with those of landslides (Figure 14a). A comparison of pixels from non-landslide riverbank areas and those affected by landslides reveals that while the spectral signatures are similar, there are noticeable differences, particularly in the CGR RGB channels. These differences, although subtle, are appreciable enough to help differentiate the two types of areas. Additionally, the lower slope values in riverbank areas, due to their gentler topography along watercourses, and reduced vegetation change indicators (NDVI Δ channels) set them apart from genuine slope failures. Post-flood sedimentation and bank erosion generate surface disturbances that resemble landslide scars, especially where floodwaters have stripped vegetation and redistributed sediments. Despite these similarities, these features were not sufficiently weighted during training on the Casola Valsenio dataset, leading to false positives in flood-affected environments. However, the distinction in the CGR RGB channels, along with the differences in slope and NIR values, played a crucial role in limiting false negatives (FN) along riverbanks, particularly in the best models (Caso 7, S7, and U7), despite the error still being present.*

*b) Geological Differences (FMA vs FAA)*

*The geological difference between Marnoso-Arenacea (FMA) and Blue Clays (FAA) proved challenging. Pixels from landslides in both lithologies were compared, with notable differences in the CGR (Red) and CGR (NIR) channels, which are critical for landslide detection (Figure 14b). FAA regions exhibit lighter colors in both pre- and post-event imagery, indicating typical erosion and sediment redistribution in badlands (see Figure 10). This aligns with the diluvial processes observed in the Blue Clays region. Training the models solely on FMA lithology, such as Casola Valsenio, rendered them ineffective when applied to FAA regions, where the spectral characteristics differ significantly. This highlights the limited transferability of models trained on a single lithology.*

*c) Plowed Fields (Modigliana)*

*Spectral signatures from landslide-affected areas in Modigliana were compared with those from plowed fields outside the landslide zones (Figure 14c). The similarities in spectral signatures, particularly in the CGR (Blue)*

*and CGR (Green) channels, led to false positives (yellow polygons) in Figure 10. Agricultural fields, when plowed, share similar characteristics with disturbed landslides, especially when considering the differences in pre-event imagery. These differences are more pronounced in the Slope and pre-event channels. However, due to the relatively low importance assigned to these channels during training on the Casola Valsenio dataset, the models struggled to differentiate these fields from landslides, resulting in confusion and false positives in plowed fields.*

*d) Shadow Effects (Predappio)*

*Shadowed regions, particularly in Predappio, made it nearly impossible for the models to classify these areas as landslides. The spectral analysis (Figure 14d) highlights much darker values in shadowed areas in the CGR (post-event) imagery, which complicates the recognition of landslides. This is due to the drastically reduced reflectance in shadowed pixels, which prevents the models from detecting landslides in these regions. Fortunately, this issue is isolated to a small portion of Predappio. The reduced illumination and altered color profiles in shadowed areas hinder the model's ability to detect the land movement, causing false negatives in these regions, as shown in Figure 10.*

[Figure]

***Figure 14****. Spectral analysis of different land types and landslide detection. **(a)** Comparison between riverbank areas affected by flooding and landslides. The blue line represents the spectral characteristics of buildings along the river, while the red line corresponds to landslides. **(b)** Spectral differences between landslides occurring on Marnoso-Arenacea Formation (FMA) and Blue Clays (FAA), showing the significant spectral dissimilarity. **(c)** Comparison between spectral signatures of cultivated fields and landslides. The green line represents cultivated fields, while the red line corresponds to landslides, with noticeable overlap in spectral characteristics. **(d)** Impact of shadowed regions on landslide detection, comparing normal conditions (red)*

*with shadowed regions (black). The graph reveals how shadows significantly reduce the ability of the models to detect landslides.*

**6.4 Future research direction**

*To overcome the limitations identified in this study, future research should focus on key areas. A primary challenge is optimizing data acquisition to improve map quality, including minimizing shadow effects by scheduling imagery collection around noon, especially during winter months. Expanding training datasets to include more plowed field examples will help models better differentiate agricultural lands from landslides, reducing false positives. Further refinement could include detailed land-use layers or slope filters to improve accuracy.*

*Improving data resolution is another important direction. While this study used 2-meter resolution data, higher-resolution datasets (e.g., 20 cm) would provide more detailed information, enabling better detection. Leveraging higher computational power would be necessary to process these datasets and enhance model performance. In terms of approach, future research could explore advanced techniques such as Multiscale Feature Pyramid Networks (FPN), which allow models to process multiple scales simultaneously, Graph Neural Networks (GNNs) for capturing complex spatial relationships in geospatial contexts, and Neural Architecture Search (NAS) to automatically identify the best network architecture, improving robustness and generalization (Lin et al., 2017; Kipf & Welling, 2017; Zoph & Le, 2017).*

*Lastly, addressing the issue of geologically diverse training datasets remains crucial. Training models on datasets with limited geological diversity can hinder generalization. Expanding datasets to include various lithologies and geomorphological features will enhance model robustness. As deep learning models evolve, it is essential to prioritize the collection of diverse, high-quality data, as even the most advanced models cannot replace the need for comprehensive datasets. Future research should focus on improving data acquisition, enhancing model generalization, and refining validation techniques to further strengthen AI-based mapping in disaster management.*

*(b) In addition, certain results, such as the inclusion of a lithological layer and the assessment of buildings potentially at risk, are introduced without having been previously described in the Methods section, which makes the paragraph difficult to follow. To improve clarity and coherence, these methodological aspects should be clearly presented earlier, and portions of the discussion should be relocated to the Results section to enhance overall readability.*

Thank you for your comment. We have revised the manuscript to improve clarity. The methodological aspects related to the inclusion of a lithological layer and the assessment of buildings at risk have now been clearly described in the Methods section. Additionally, relevant portions of the discussion have been moved to the Results section for better coherence and readability.

**"4.8 Identification of At-Risk Buildings**

*In the context of emergency mapping, a key objective was to identify buildings at risk due to the landslides triggered by the May 2023 event. This was a critical task for the Civil Protection as it allowed for timely intervention and damage assessment. Given the extensive spatial distribution of the damage caused by the event, it was essential to develop a systematic method to detect and assess the buildings at risk. Additionally, accurate damage estimation was necessary to request financial support under the European Union Solidarity Fund (EUSF), adhering to the EU's required timelines for funding requests.*

*To evaluate the effectiveness of our automated landslide mapping models in identifying at-risk buildings, we created a 20-meter buffer around the predicted landslide boundaries. This buffer was compared to the list of buildings that the Civil Protection had classified as "at risk," based on manual mapping (Pizziolo et al., 2024; Berti et al., 2025). The buildings identified by the automated models were then compared to these reference buildings, allowing us to assess the models' capacity for accurate identification. The confusion matrix was used to quantify the performance of the models in identifying buildings at risk, including both false positives (incorrectly flagged buildings) and false negatives (missed buildings).*

*This methodology ensured that the models were tested in the real-world context of emergency response, where timely and accurate damage assessments are crucial for effective resource allocation and intervention. The results of this methodology are discussed further in Section 6.3, while the quantitative outcomes of our models are presented in Section 5.3."*

*"5.3 Identification of At-Risk Buildings*

*The identification of at-risk buildings is a crucial component in emergency response scenarios. In the aftermath of the May 2023 disaster, buildings located within landslide boundaries or within a 20-meter buffer zone were considered at risk, influencing subsequent evaluations carried out by the relevant authorities. To assess the effectiveness of our automated landslide mapping for this purpose, we compared the buildings identified by our models to those derived from manual mapping (Pizziolo et al., 2024; Berti et al., 2025).*

*Figure 12 presents the confusion matrix for the 14 model configurations, showing the comparison between the buildings flagged by our models and those mapped manually. The F1-scores obtained for each case highlight the models' ability to identify structures at risk. For Case 1, U-Net produced an F1-score of 0.53, while SegFormer performed slightly better at 0.67. The highest F1-scores were achieved in Case 7, with U-Net reaching 0.79 and SegFormer 0.75. Notably, models U2, U4, S6, and U7 consistently yielded the best results, with F1-scores averaging around 0.78, demonstrating their effectiveness in accurately identifying buildings at risk.*

*Despite the relatively high F1-scores, the models still produced both false positives (incorrectly flagged buildings) and false negatives (missed buildings), which underscores the necessity of manual verification in scenarios where precision is critical. However, the performance of the automated models is considerably higher than the manual mapping, especially when considering the large-scale, time-sensitive nature of emergency response. These results suggest that the automated models can significantly assist in the rapid identification of at-risk structures, potentially saving valuable time in the early stages of disaster response. These results are discussed in Section 6.3.*

[Figure]

***Figure 12.*** *Comparison of buildings identified as at risk (within landslide boundaries or within a 20-meter buffer) by automated mapping methods and manual mapping (ground truth), showing the confusion matrices for all five cases evaluated."*

*"6.3 Identification of At-Risk Buildings*

*A further consideration in our study is the use of automated landslide maps to identify damage and at-risk structures. Following the May 2023 disaster, the Emergency Commission determined that all buildings located within landslide boundaries or within a 20-meter buffer were considered at risk and potentially subject to relocation. To evaluate the effectiveness of our automated mapping products for this task, we compared the buildings identified automatically with those derived from manual mapping.*

*The F1-scores for the models indicate that "U2", "U4", "S6", and "U7" are the most accurate CNN outputs, identifying on average 528 out of 654 buildings at risk (~0.78 F1-score). These results demonstrate the models' ability to estimate the spatial extent of the phenomenon, even when using Sentinel-2 imagery at 10 m resolution (U2). However, all models produced both false positives (incorrectly flagged buildings) and false negatives (missed buildings), underscoring the need for manual verification when high accuracy is required. Even a small number of misclassified structures, especially inhabited buildings, can have serious consequences in emergency situations, thus highlighting the importance of improving model performance for more effective response efforts.*

*Improving the landslide mapping itself is crucial for improving the accuracy of at-risk building identification. As discussed in Section 7, refining mapping methodologies will lead to more reliable and consistent results in future disaster scenarios.*

***Minor Review***

***R1.9 – "Abstract"***

**Abstract**: *Well written but slightly long. Can be more objective and tightened by removing methodological repetition.*

We thank the reviewer for this helpful suggestion and agree that the Abstract could be streamlined. We have revised the Abstract to improve conciseness and objectivity by removing repetitive methodological details (e.g., extended descriptions of training/testing setup and input-layer combinations) and by focusing more directly on the key findings and their implications for rapid emergency mapping. The revised version retains the core study motivation, experimental setting, and main outcomes while reducing length and improving readability.

*"The catastrophic rainfall events of May 2023 in the Emilia-Romagna region (Italy) triggered more than 80,000 landslides, documented in the RER2023 inventory (Berti et al., 2025) and released as an open-access dataset (Pizziolo et al., 2024; https://doi.org/10.5281/zenodo.13742643). Rapid landslide mapping was a critical emergency task, yet manual delineation required substantial time and resources (Berti et al., 2025). Building on previous automated mapping tests in Casola Valsenio (Berti et al., 2026), we evaluate the potential of deep-learning segmentation to support rapid disaster response by comparing U-Net and SegFormer under realistic constraints, including limited training data and heterogeneous test conditions across four municipalities.*

*Both models produced usable landslide maps, with F1 and IoU scores indicating comparable overall performance. SegFormer showed greater stability across input-data scenarios and geological settings, whereas U-Net exhibited larger performance variability and achieved the highest scores when richer inputs were available. Errors were concentrated in shadowed areas, cultivated fields, and lithologically distinct terrains. Performance was lowest in the Brisighella FAA sector (Blue Clay formations), indicating limited*

*generalization when the training data lack lithological diversity and supporting the need for geologically representative training datasets.*

*Overall, the results support automated mapping as a first-pass product to accelerate emergency response, enabling rapid screening and prioritization for subsequent expert validation. While manual revision remains necessary for high-stakes applications, deep-learning-based mapping provides a scalable approach to deliver timely and spatially detailed hazard information during large events."*

**R1.10 – "3. Study Area"**

*(a) 136-137: Readability would be improved by explicitly citing the name of the municipalities;*

Thank you for the suggestion. The sentence has been revised to explicitly list the municipalities analysed: Casola Valsenio, Brisighella, Modigliana, and Predappio.

*"Automated methods for landslide mapping were applied in the four municipalities shown in Figure 2 (Casola Valsenio, Brisighella, Modigliana and Predappio)."*

*(b) 144: Text states that "about 25% of the recorded 80,997 landslides events", adding the quantity of landslides would also improve the comprehension of the impacted area;*

Agreed. The text has been updated to report both the percentage and the absolute number of landslides (20,148 events), improving clarity on the extent of the impacted areas.

*"The other three areas were selected because together they encompass approximately 35% of the regions most impacted by landslides and represent around 25% of the recorded 80,997 landslide events (20148)."*

*(c) Figure 2 - Increase the legend size*

We agree with the Reviewer. The legend size in Figure 2 has been increased to improve readability and facilitate interpretation of the mapped information.

[Figure]

*(d) 193: Correct from "sandstone-rice" to "sandstone-rich".*

Corrected as suggested.

*"The northern sector is distinguished by predominantly sandstone-rich layers (A/P > 3) along with"*

***R1.11 – "3. Methods"***

*(a) Review the numbering of the sections*

Corrected. Thank you for pointing out this oversight.

*(b) Figure 4: Sentinel images are with low contrast which difficult the visualization.*

We acknowledge this point. Sentinel-2 images are displayed using their original radiometric values to reflect the actual data used for model training and inference. Improving contrast would result in a visualization inconsistent with the input provided to the algorithms. Low contrast is a common limitation in emergency scenarios and does not affect model performance, as deep-learning models operate on radiometric information rather than visual interpretability.

*(c) 304-309 - Should also describe the product level that was used in the study.*

Agreed. The text has been revised to explicitly state that Sentinel-2 Level-2A (L2A, bottom-of-atmosphere surface reflectance) products were used for both pre- and post-event imagery.

"

- *L1)* *Post-event Sentinel-2 images**: Four-band (RGB+NIR) satellite images at 10 m spatial resolution, acquired after the second rainfall event on May 23, 2023. Sentinel-2 Level-2A (L2A) products (bottom-of-atmosphere surface reflectance) were used. They represent the first post-event Sentinel-2 images obtained with minimal cloud cover (Fig. 4b).*
- *L2)* *Pre-event Sentinel-2 images**: Four-band (RGB+NIR) satellite images at 10 m spatial resolution acquired in May 2022, one year prior to the event. Sentinel-2 Level-2A (L2A) products (bottom-of-atmosphere surface reflectance) were used. These images provide a similar vegetation state to that of May 2023 and were selected due to the lack of cloud-free images in the two months preceding the event (Fig. 4a)."*

*(d) 320-354 - Text refers to cases (e.g case 1, case 2 ..) however this definition is not represented or cited in table 2.*

Thank you for the clarification. Tables 3 and 5 have been revised to explicitly include and reference the input-layer cases (Cases 1–7).

*"Table 3. Overview of the various input layer configurations used during the training processes. 'NDVI' refers to the 'ΔNDVI-CGR' Change map created using CGR imagery, while 'ΔNDVI-S2' refers to the NDVI Change map created using Sentinel-2 imagery.*

| Cases | Name | Model | Sentinel2 post [L1] | Sentinel2 pre [L2] | Sentinel2 ΔNDVI [L3] | CGR post [L4] | AGEA pre [L5] | CGR ΔNDVI [L6] | Slope [L7] |
|---|---|---|---|---|---|---|---|---|---|
| Case 1 | U1 | U-Net | ✓ | X | X | X | X | X | ✓ |
|  | S1 | SegForm | ✓ | X | X | X | X | X | ✓ |
| Case 2 | U2 | U-Net | ✓ | ✓ | ✓ | X | X | X | ✓ |
|  | S2 | SegForm | ✓ | ✓ | ✓ | X | X | X | ✓ |
| Case 3 | U3 | U-Net | X | X | X | ✓ | X | X | ✓ |
|  | S3 | SegForm | X | X | X | ✓ | X | X | ✓ |
| Case 4 | U4 | U-Net | ✓ | ✓ | ✓ | ✓ | X | X | ✓ |
|  | S4 | SegForm | ✓ | ✓ | ✓ | ✓ | X | X | ✓ |
| Case 5 | U5 | U-Net | X | X | X | ✓ | ✓ | X | ✓ |
|  | S5 | SegForm | X | X | X | ✓ | ✓ | X | ✓ |
| Case 6 | U6 | U-Net | X | X | X | ✓ | ✓ | ✓ | ✓ |
|  | S6 | SegForm | X | X | X | ✓ | ✓ | ✓ | ✓ |
| Case 7 | U7 | U-Net | ✓ | ✓ | ✓ | ✓ | ✓ | ✓ | ✓ |
|  | S7 | SegForm | ✓ | ✓ | ✓ | ✓ | ✓ | ✓ | ✓ |

**Table 5.** *F1-score and Intersection over Union (IoU) obtained by the two deep-learning semantic segmentation models (U = U-Net; S = SegFormer) for the seven input-layer configurations (cases 1–7; Table 3), evaluated against the manually mapped reference inventory. Models were trained in Casola Valsenio and applied without retraining to the external test municipalities (Brisighella FMA, and Brisighella FAA, Modigliana, Predappio). The table therefore reports 56 model outputs (7 cases × 4 municipalities × 2 models)."*

| Cases | Models' Name | Casola Valsenio | | Brisighella FMA | | Brisighella FAA | | Modigliana | | Predappio | |
|---|---|---|---|---|---|---|---|---|---|---|---|
| | | F1 | IoU | F1 | IoU | F1 | IoU | F1 | IoU | F1 | IoU |
| Case 1 | U1 | 0.56 | 0.39 | 0.50 | 0.33 | 0.32 | 0.19 | 0.48 | 0.31 | 0.42 | 0.27 |
| | S1 | 0.47 | 0.31 | 0.46 | 0.30 | 0.34 | 0.21 | 0.45 | 0.29 | 0.46 | 0.30 |
| Case 2 | U2 | 0.64 | 0.47 | 0.57 | 0.40 | 0.38 | 0.24 | 0.55 | 0.38 | 0.52 | 0.35 |
| | S2 | 0.62 | 0.45 | 0.57 | 0.40 | 0.47 | 0.31 | 0.57 | 0.40 | 0.54 | 0.37 |
| Case 3 | U3 | 0.70 | 0.54 | 0.55 | 0.38 | 0.29 | 0.17 | 0.45 | 0.29 | 0.39 | 0.24 |
| | S3 | 0.54 | 0.37 | 0.49 | 0.32 | 0.32 | 0.19 | 0.44 | 0.29 | 0.44 | 0.28 |
| Case 4 | U4 | 0.72 | 0.56 | 0.62 | 0.45 | 0.47 | 0.31 | 0.58 | 0.41 | 0.55 | 0.38 |
| | S4 | 0.68 | 0.52 | 0.62 | 0.45 | 0.49 | 0.33 | 0.60 | 0.43 | 0.59 | 0.42 |
| Case 5 | U5 | 0.71 | 0.55 | 0.62 | 0.45 | 0.50 | 0.34 | 0.56 | 0.39 | 0.53 | 0.36 |
| | S5 | 0.65 | 0.48 | 0.60 | 0.43 | 0.43 | 0.28 | 0.56 | 0.39 | 0.56 | 0.39 |
| Case 6 | U6 | 0.72 | 0.56 | 0.63 | 0.46 | 0.53 | 0.36 | 0.59 | 0.42 | 0.56 | 0.38 |
| | S6 | 0.65 | 0.48 | 0.58 | 0.41 | 0.41 | 0.26 | 0.54 | 0.37 | 0.53 | 0.36 |
| Case 7 | U7 | 0.73 | 0.57 | 0.63 | 0.46 | 0.53 | 0.36 | 0.60 | 0.42 | 0.56 | 0.39 |
| | S7 | 0.66 | 0.49 | 0.61 | 0.44 | 0.52 | 0.35 | 0.60 | 0.43 | 0.60 | 0.43 |

**R1.12 – "3.7 Evaluation Metrics and Expert Judgment "**

*Lines 498-501 where the F1 metric is described, the text says that the metric is useful for imbalanced problems, but this was not clearly discussed in the methodology, was the data used in the experiment imbalanced?*

Thank you for highlighting this point. The class imbalance has now been explicitly quantified in Section 4.4, where pixel-level imbalance ratios are reported for the training and test areas. Section 4.7.1 has been revised accordingly to justify the use of the F1-score in light of this imbalance.

*"4.4 Data preparation and training-testing design*

*…*

*Within Casola Valsenio, a stratified random sampling strategy was used to partition the tiles into training, validation, and internal testing subsets. Out of 97 tiles, 60 (62%) were assigned to training, 15 (15%) to validation, and 22 (23%) to internal testing (Fig. 5). External evaluation was then conducted on all tiles from the three remaining municipalities, providing a spatial hold-out test to assess cross-municipality*

*generalisation. This tile-based spatial splitting yields a strongly imbalanced pixel-level segmentation problem, because landslide pixels represent only a small fraction of the mapped area. In Casola Valsenio (training area), landslide pixels account for ~ 4% of the total pixels (non-landslide:landslide ratio ~24:1). A comparable imbalance characterises the external test areas, with ~6% landslide pixels in Predappio (~ 16:1), ~7.6% in Modigliana (~12:1), and ~5% in Brisighella FMA (~19:1). The most extreme imbalance occurs in Brisighella FAA, where landslide pixels represent ~2% of the pixels (~48:1). The implications of this class imbalance for model evaluation are discussed in Section 4.7.*

*...*

*4.7 Evaluation Metrics and Expert Judgment*

*4.7.1 Quantitative Metrics*

*...*

*The F1 score (Dice coefficient) is the harmonic mean of precision and recall (Tharwat, 2020). We report F1 alongside IoU because the pixel-wise segmentation task is strongly imbalanced (Section 4.4), with landslide pixels representing only a small fraction of the mapped area. The F1 score is calculated as:"*

**R1.12 – "5 Results"**

*(a) First paragraph (556-562) could be adapted and turned into the legend of table 4, as it does not describe any result, this would improve readability.*

Following the major revisions, this issue has been addressed. The Results section has been rewritten to focus on quantitative findings, and the caption of Table 5 has been expanded to include descriptive information previously contained in the introductory paragraph.

*"**5. Results***

*5.1 Model results and performance*

*The results of the analysis are reported in Table 5, which summarizes the F1 and IoU scores obtained by comparing the automated landslide maps generated by the two deep-learning models (U = U-Net; S = SegFormer) with the manually mapped reference inventory. Seven combinations of input layers (cases 1-7; Table 1) were tested, representing progressively richer input information. The models were trained using data from Casola Valsenio and subsequently applied to four independent test areas: Brisighella FMA, Brisighella FAA, Modigliana and Predappio. In total, 56 automated landslide maps were produced (7 input configurations × 4 test areas × 2 models).*

*Across all municipalities, the highest performance was obtained by the most information-rich configurations, particularly U7 and U6 for U-Net and S7 for SegFormer. In Casola Valsenio, the best configuration reached F1 = 0.73 and IoU = 0.57 (U7), while in Brisighella FMA, Brisighella FAA, Modigliana and Predappio the highest F1-scores were consistently around 0.60–0.63 (e.g., U7 = 0.56–0.63; S7 = 0.60–0.61). Performance was systematically lower in Brisighella FAA, where the best configurations reached F1 = 0.53 and IoU = 0.36 (U6/U7), and F1 = 0.52 and IoU = 0.35 (S7) (Table 5).*

*Despite relying on a reduced set of inputs, the Sentinel-2-based configuration S2 (post-event Sentinel-2 plus Sentinel-2 NDVI change) achieved competitive results across municipalities. For instance, in Casola Valsenio S2 reached F1 = 0.62 and IoU = 0.45, compared with F1 = 0.73 and IoU = 0.57 for the best-performing configuration (U7). In Brisighella FMA, Modigliana and Predappio, S2 yielded F1-scores between 0.54 and 0.57 (IoU = 0.37–0.40), which are close to the values obtained by the strongest configurations (typically within ~0.03–0.06 in F1 and ~0.03–0.06 in IoU). In Brisighella FAA, S2 remained among the most stable configurations (F1 = 0.47; IoU = 0.31), whereas some intermediate cases showed marked drops (e.g., U3: F1 = 0.29; IoU = 0.17).*

*Overall, increasing the number of input layers resulted in measurable but limited performance improvements. The largest gain was associated with the inclusion of NDVI change information (case 2 vs. case 1), while subsequent additions of high-resolution inputs generally produced smaller incremental changes. These results indicate that streamlined configurations based on Sentinel-2 data (case 2), optionally complemented by high-resolution imagery, can achieve segmentation performance close to that obtained using the full set of available inputs (Table 5)."*

*"**Table 5.** F1-score and Intersection over Union (IoU) obtained by the two deep-learning semantic segmentation models (U = U-Net; S = SegFormer) for the seven input-layer configurations (cases 1–7; Table 3), evaluated against the manually mapped reference inventory. Models were trained in Casola Valsenio and applied without retraining to the external test municipalities (Predappio, Modigliana, Brisighella FMA, and Brisighella FAA). The table therefore reports 56 model outputs (7 cases × 4 municipalities × 2 models)."*

*(b) Figure 7 - Legend states "case 1 - 7" but in the table it is referred as "U1", "S1", "U2", "S2", also inputting these names would facilitate the comprehension of the table. The size of the legends should be increased.*

The figure caption has been revised to explicitly clarify the correspondence between Cases 1-7 and labels U1–U7 and S1–S7. The legend size has also been increased to improve readability.

[Figure]

*"**Figure 8**. F1-score comparison across the seven input-layer configurations for U-Net (left) and SegFormer (right). The numbering matches the input-layer cases defined in Table 3 (e.g., U1 and S1 refer to Case 1; U2 and S2 refer to Case 2), and values are summarised in Table 5."*

*(c) Figure 8 and 9 - Changing the colors used to highlight the TP and FP would improve the visualization.*

Figures 9 and 10 have been revised by darkening the mask overlays and increasing contrast, improving the visual separation between the inventory and the model predictions.

[Figure]

*"**Figure 9.** Comparative analysis of landslide mapping results using the U7 (U-Net) and S7 (SegFormer) configurations across the analysed municipalities (Casola Valsenio, Brisighella FMA, Brisighella FAA, Modigliana and Predappio). The figure shows the spatial distribution and extent of the mapped landslides for both models. In the overlays, the reference inventory (True map) is shown in blue and the model prediction in yellow; their intersection is shown in brown (true positives), while blue-only and yellow-only areas correspond to false negatives and false positives, respectively."*

[Figure]

*"**Figure 10.** Representative examples of challenging scenarios for landslide detection using the U7 (U-Net) and S7 (SegFormer) configurations across municipalities. The panels highlight recurrent error patterns under various environmental conditions: Brisighella FMA shows riverbanks and buildings as False Positives (FP), Brisighella FAA displays both False Positives and False Negatives (FP & FN) in badlands, Modigliana FN is observed on plowed fields, and Predappio FN occurs under shadows."*

*(d) Paragraphs from 628-655 highlight difficulties of the model in the segmentation over different areas, but figure 9 does not highlight those areas making it harder to follow the paragraph.*

We agree with the reviewer. The Results section has been revised to explicitly link the discussed error patterns to the corresponding panels in Figures 9 and 10. The captions were updated to clarify the visual interpretation of false positives, false negatives, and true positives, and the text now follows the same spatial order as the figures, improving coherence and readability.

*"Figures 9 and 10 complement the quantitative results by providing a visual interpretation of model performance under contrasting conditions. Figure 9 illustrates representative cases in which both models*

*performed well, showing a strong spatial agreement between the manually mapped inventory and the automated outputs. In these panels, the overlap between the reference landslide map (blue) and the predicted map (yellow) produces a beige colour, corresponding to true positives and indicating accurate landslide delineation. These examples mainly refer to debris slides and debris flows occurring in well-illuminated areas with widespread vegetation removal, where spectral and textural changes are pronounced.*

*Figure 10, by contrast, focuses on more challenging scenarios, where the models exhibited recurrent error patterns. In Brisighella FMA, false positives (yellow polygons) are frequently mapped along riverbanks and on recently constructed buildings, likely associated with flood-related sediment redistribution and strong spectral contrasts. The most critical conditions are observed in Brisighella FAA, where widespread false positives occur within badland morphologies developed on Blue Clay formations, highlighting the limited transferability of models trained predominantly on flysch-dominated terrains. In Modigliana, false positives (yellow polygons) are often observed in cultivated fields, where agricultural disturbance produces spectral signatures similar to recent landslides. Finally, in Predappio, false negatives (blue polygons) dominate in shadowed slopes, where reduced illumination and altered colour profiles limit the detectability of landslide features.*

*By explicitly contrasting successful detections (Figure 9) with failure modes (Figure 10), these visual examples clarify how different surface conditions, illumination effects, and lithological settings influence model performance, supporting the quantitative trends reported in Table 5 and Figure 8.*

**Figure 9**. *Comparative analysis of landslide mapping results using the U7 (U-Net) and S7 (SegFormer) configurations across the analysed municipalities (Casola Valsenio, Brisighella FMA, Brisighella FAA, Modigliana and Predappio). The figure shows the spatial distribution and extent of the mapped landslides for both models. In the overlays, the reference inventory (True map) is shown in blue and the model prediction in yellow; their intersection is shown in brown (true positives), while blue-only and yellow-only areas correspond to false negatives and false positives, respectively.*

**Figure 10.** *Representative examples of challenging scenarios for landslide detection using the U7 (U-Net) and S7 (SegFormer) configurations across municipalities. The panels highlight recurrent error patterns under various environmental conditions: Brisighella FMA shows riverbanks and buildings as False Positives (FP), Brisighella FAA displays both False Positives and False Negatives (FP & FN) in badlands, Modigliana FN is observed on plowed fields, and Predappio FN occurs under shadows."*

**R1.13 – "6 Discussion"**

*Paragraph from 715-726 states that another model was trained to evaluate the impact of including lithology but no explanation was added to describe how this was done. The categorical layer was converted to dummy layers (one for each category) or was added as a categorical layer? Extra information would improve the comprehension*

Thank you for this observation. We have clarified the interpretation of the results by specifically addressing the spectral response of the mapped targets. In the analysis, we have detailed how the spectral characteristics of different areas, such as riverbanks, landslides, cultivated fields, and shadowed regions, were compared. The differences in the spectral signatures, especially in the CGR RGB, slope, and NIR channels, were explicitly discussed. This explanation has been incorporated to provide a better understanding of how these spectral responses influence the model's performance and detection capabilities. The revised text now includes these interpretations to make the results more comprehensible in terms of the spectral response of the mapped targets.

*"To further investigate the role of geology in model generalization, an additional analysis was conducted to assess the impact of including lithology among the input layers. For this, the lithological layer was categorized*

*into 8 distinct classes, with each class corresponding to a unique Unit ID (as detailed in Table 2). These classes were numerically encoded from 1 to 8, representing different lithological units. We compared the performance of Case S3 (RGB + slope) with the same configuration plus the lithology layer. However, this did not yield significant improvements in Casola, Predappio, Modigliana, or Brisighella MA, as these areas are predominantly characterized by the Marnoso-Arenacea Formation ("Unit 7"), which was already well represented in the training dataset (Figure 2a). As a result, the addition of the lithology layer provided redundant information. However, when the same model was applied to Brisighella FAA, where the lithology is dominated by Blue Clays ("Unit 1"), the model failed to detect any landslides, resulting in a very low F1-score of 0.02. This underscores a key limitation: simply adding lithology as a static input layer is insufficient for ensuring generalization. If the training dataset is not lithologically balanced, the model may reinforce existing biases, associating landslide occurrence primarily with "Unit 7" and failing to detect events in "Unit 1".*

*This finding highlights a fundamental issue in landslide detection models: effectively incorporating lithological information requires more than just adding it as an input feature. The training dataset itself must include landslide polygons from a representative range of lithological settings. When the model was explicitly trained and tested on Brisighella FAA using the same configuration (Case S3: RGB + slope), the F1-score increased from 0.32 to 0.41, confirming that geological consistency in the training data significantly improves the model's ability to detect landslides in previously underrepresented domains."*

*The discussion in lines 734–741 regarding the F1-score relies on comparisons with generic machine-learning benchmarks rather than on metrics reported in landslide-mapping studies. The analysis should instead be grounded in the context of landslide detection and segmentation literature. In particular, the manuscript should address why F1-scores in landslide mapping are often relatively low, discussing contributing factors such as class imbalance between landslide and non-landslide areas, the spatial heterogeneity of landslides, uncertainties in reference inventories, and the influence of image resolution and labeling quality.*

Thank you for your comment. We have revised the discussion to address the F1-score in the context of landslide mapping. While the F1-scores in this study may seem modest compared to machine learning benchmarks (e.g., Ghorbanzadeh et al., 2021; Prakash et al., 2021), such scores are typically lower in landslide detection due to factors like class imbalance, spatial heterogeneity, uncertainties in reference inventories, and image resolution. Our dataset, designed for emergency response, maps a small portion of the affected area (Casola Valsenio), resulting in a train-validation-test ratio of 12% for training, 3% for validation, and 85% for testing. These ratios and the task complexity contribute to the modest scores, which are typical in remote sensing applications requiring high-resolution, fine-grained feature extraction.

*"Although the F1-scores and Intersection over Union (IoU) achieved in this study may appear modest compared to generic machine-learning benchmarks (e.g., Ghorbanzadeh et al., 2021, 2022; Prakash et al., 2021; Meena et al., 2023), it is important to ground these results within the context of landslide detection and segmentation literature, where lower scores are often observed. Several factors contribute to these relatively low F1-scores, including class imbalance between landslide and non-landslide areas, the spatial heterogeneity of landslides, uncertainties in reference inventories, and the influence of image resolution and labeling quality. Landslide mapping, unlike conventional image classification tasks, requires precise delineation of irregular and complex shapes across highly variable terrain, which makes it more challenging to achieve high scores.*

*Our dataset was specifically designed for emergency response scenarios, where a small portion of the affected area (Casola Valsenio) is mapped and the model is applied to the rest. This led to a train-validation-test ratio of 12% for training, 3% for validation, and 85% for testing. The limited training and validation sets, combined with the complex nature of landslide mapping, contribute to modest F1-scores. Such a trade-off between accuracy and generalization is common in remote sensing, where high-resolution, fine-grained feature extraction is both critical and difficult to achieve at a large scale.*

*Nevertheless, the potential of these automated approaches in emergency scenarios is considerable. By rapidly generating landslide maps of comparable quality to expert-drawn products, these methods could significantly accelerate initial response efforts. Reducing the need for time-consuming manual digitization could save weeks of work, which is crucial during crisis events (Berti et al., 2025). The automatically generated maps may serve as a robust initial product, enabling practitioners to focus on refinement and validation rather than starting from scratch, ultimately delivering high-quality final maps in a fraction of the time required for full manual mapping."*

**R1.14 – "7 Conclusion"**

*This section should also address future research directions aimed at overcoming the limitations identified in the study.*

We thank the reviewer for this suggestion. We have explicitly addresses future research directions aimed at overcoming the limitations identified in the study. Additionally, we conclude by emphasizing that data quality and diversity are more critical than model architecture sophistication, highlighting a key direction for future research in this field.

*"6.4 Future Research Directions*

*To overcome the limitations identified in this study, future research should focus on key areas. A primary challenge is optimizing data acquisition to improve map quality, including minimizing shadow effects by scheduling imagery collection around noon, especially during winter months. Expanding training datasets to include more plowed field examples will help models better differentiate agricultural lands from landslides, reducing false positives. Further refinement could include detailed land-use layers or slope filters to improve accuracy.*

*Improving data resolution is another important direction. While this study used 2-meter resolution data, higher-resolution datasets (e.g., 20 cm) would provide more detailed information, enabling better detection. Leveraging higher computational power would be necessary to process these datasets and enhance model performance. In terms of approach, future research could explore advanced techniques such as Multiscale Feature Pyramid Networks (FPN), which allow models to process multiple scales simultaneously, Graph Neural Networks (GNNs) for capturing complex spatial relationships in geospatial contexts, and Neural Architecture Search (NAS) to automatically identify the best network architecture, improving robustness and generalization (Lin et al., 2017; Kipf & Welling, 2017; Zoph & Le, 2017).*

*Lastly, addressing the issue of geologically diverse training datasets remains crucial. Training models on datasets with limited geological diversity can hinder generalization. Expanding datasets to include various lithologies and geomorphological features will enhance model robustness. As deep learning models evolve, it is essential to prioritize the collection of diverse, high-quality data, as even the most advanced models cannot replace the need for comprehensive datasets. Future research should focus on improving data acquisition, enhancing model generalization, and refining validation techniques to further strengthen AI-based mapping in disaster management.*

**7. Conclusion**

*This study investigated the potential of automated landslide mapping to support rapid emergency response following the extreme meteorological events of May 2023 in Emilia-Romagna, Italy. Using a deep learning approach, we trained and tested two semantic segmentation models, U-Net and SegFormer, on high-resolution aerial imagery, Sentinel-2 data, NDVI change maps, and slope data. The training was conducted solely in the municipality of Casola Valsenio, while model performance was assessed on three additional municipalities (Brisighella, Modigliana, Predappio), chosen for their geological settings and significant landslide occurrence.*

*In the training area of Casola Valsenio, both U-Net and SegFormer achieved high performance, with consistent results across different input combinations. The models showed no significant variation in performance based on the input layers, as long as high-resolution post-event imagery (CGR) was included. However, when applied to external regions such as Brisighella, Modigliana, and Predappio, the models experienced a decrease in generalization. Despite this, the automated mapping still successfully identified the majority of landslides, demonstrating its utility in emergency scenarios.*

*In areas like Brisighella FAA (Blue Clays), where relevant lithologies were underrepresented in the training set, the models struggled to generalize effectively. This highlights the importance of incorporating a diverse range of lithologies in the training data to ensure robust model generalization across different geologically complex regions.*

*The choice of model architecture, whether U-Net or SegFormer, had minimal impact on performance, with both models showing similar results. Performance decreased with less detailed inputs, like Sentinel imagery alone, and improved with more rich inputs, reinforcing that data quality, not architecture, drives performance.*

*In emergency contexts, where external data for validation may not be available, expert judgment becomes essential for selecting the most reliable maps when ground truth data is lacking. While automated maps offer rapid assessments, they often require expert adjustments to refine outputs offering a balanced approach between speed and accuracy. The correct and thoughtful integration of AI-based systems into civil protection protocols represents a critical step forward, ensuring that these technologies complement existing procedures and improve overall response effectiveness."*

*"References*

*Lin, T.-Y., Dollár, P., Girshick, R., He, K., & Hariharan, B. (2017). Feature Pyramid Networks for Object Detection. CVPR 2017. https://arxiv.org/abs/1612.03144*

*Kipf, T. N., & Welling, M. (2017). Semi-Supervised Classification with Graph Convolutional Networks. ICLR 2017. https://arxiv.org/abs/1609.02907*

*Zoph, B., & Le, Q. V. (2017). Neural Architecture Search with Reinforcement Learning. ICLR 2017. https://arxiv.org/abs/1611.01578"*